# Variational Bayesian Unlearning

**Quoc Phong Nguyen, Bryan Kian Hsiang Low, and Patrick Jaillet**[†]
Dept. of Computer Science, National University of Singapore, Republic of Singapore
Dept. of Electrical Engineering and Computer Science, MIT, USA[†]
{qphong,lowkh}@comp.nus.edu.sg, jaillet@mit.edu[†]

## Abstract

This paper studies the problem of approximately unlearning a Bayesian model from a small subset of the training data to be erased. We frame this problem as one of minimizing the Kullback-Leibler divergence between the approximate posterior belief of model parameters after directly unlearning from erased data vs. the exact posterior belief from retraining with remaining data. Using the *variational inference* (VI) framework, we show that it is equivalent to minimizing an evidence upper bound which trades off between fully unlearning from erased data vs. not entirely forgetting the posterior belief given the full data (i.e., including the remaining data); the latter prevents catastrophic unlearning that can render the model useless. In model training with VI, only an approximate (instead of exact) posterior belief given the full data can be obtained, which makes unlearning even more challenging. We propose two novel tricks to tackle this challenge. We empirically demonstrate our unlearning methods on Bayesian models such as sparse Gaussian process and logistic regression using synthetic and real-world datasets.

## 1 Introduction

Our interactions with *machine learning* (ML) applications have surged in recent years such that large quantities of users' data are now deeply ingrained into the ML models being trained for these applications. This greatly complicates the regulation of access to each user's data or implementation of *personal data ownership*, which are enforced by the General Data Protection Regulation in the European Union [24]. In particular, if a user would like to exercise her *right to be forgotten* [24] (e.g., when quitting an ML application), then it would be desirable to have the trained ML model "unlearn" from her data. Such a problem of *machine unlearning* [4] extends to the practical scenario where a small subset of data previously used for training is later identified as malicious (e.g., anomalies) [4, 9] and the trained ML model can perform well once again if it can unlearn from the malicious data.

A naive alternative to machine unlearning is to simply retrain an ML model from scratch with the data *remaining* after *erasing* that to be unlearned from. In practice, this is prohibitively expensive in terms of time and space costs since the remaining data is often large such as in the above scenarios. How then can a trained ML model *directly* and efficiently unlearn from a small subset of data to be erased to become (a) exactly and if not, (b) approximately close to that from retraining with the large remaining data? Unfortunately, (a) exact unlearning is only possible for selected ML models (e.g., naive Bayes classifier, linear regression, $k$-means clustering, and item-item collaborative filtering [4, 12, 30]). This motivates the need to consider (b) approximate unlearning as it is applicable to a broader family of ML models like neural networks [9, 13] but, depending on its choice of loss function, may suffer from *catastrophic unlearning*[1] that can render the model useless. For example, to mitigate this issue, the works of [9, 13] have to "patch up" their loss functions by additionally bounding the loss incurred

by erased data with a rectified linear unit and injecting a regularization term to retain information of the remaining data, respectively. This begs the question whether there exists a loss function that can *directly* quantify the approximation gap and *naturally* prevent catastrophic unlearning.

Our work here addresses the above question by focusing on the family of Bayesian models. Specifically, our proposed loss function measures the *Kullback-Leibler* (KL) divergence between the approximate posterior belief of model parameters by directly unlearning from erased data vs. the exact posterior belief from retraining with remaining data. Using the *variational inference* (VI) framework, we show that minimizing this KL divergence is equivalent to *minimizing* (instead of maximizing) a counterpart of the evidence lower bound called the *evidence upper bound* (EUBO) (Sec. 3.2). Interestingly, the EUBO lends itself to a natural interpretation of a trade-off between fully unlearning from erased data vs. not entirely forgetting the posterior belief given the *full* data (i.e., including the remaining data); the latter prevents catastrophic unlearning induced by the former.

Often, in model training, only an approximate (instead of exact) posterior belief of model parameters given the full data can be learned, say, also using VI. This makes unlearning even more challenging. To tackle this challenge, we analyse two sources of inaccuracy in the approximate posterior belief learned using VI, which lay the groundwork for proposing our first trick of an *adjusted likelihood* of erased data (Sec. 3.3.1): Our key idea is to curb unlearning in the region of model parameters with low approximate posterior belief where both sources of inaccuracy primarily occur. Additionally, to avoid the risk of incorrectly tuning the adjusted likelihood, we propose another trick of *reverse KL* (Sec. 3.3.2) which is naturally more protected from such inaccuracy without needing the adjusted likelihood. Nonetheless, our adjusted likelihood is general enough to be applied to reverse KL.

VI is a popular approximate Bayesian inference framework due to its scalability to massive datasets [15, 18] and its ability to model complex posterior beliefs using generative adversarial networks [33] and normalizing flows [21, 29]. Our work in this paper exploits VI to broaden the family of ML models that can be unlearned, which we empirically demonstrate using synthetic and real-world datasets on several Bayesian models such as sparse Gaussian process and logistic regression with the approximate posterior belief modeled by a normalizing flow (Sec. 4).

## 2   Variational Inference (VI)

In this section, we revisit the VI framework [2] for learning an approximate posterior belief of the parameters $\boldsymbol{\theta}$ of a Bayesian model. Given a prior belief $p(\boldsymbol{\theta})$ of the unknown model parameters $\boldsymbol{\theta}$ and a set $\mathcal{D}$ of training data, an approximate posterior belief $q(\boldsymbol{\theta}|\mathcal{D}) \approx p(\boldsymbol{\theta}|\mathcal{D})$ is being optimized by minimizing the KL divergence $\mathrm{KL}[q(\boldsymbol{\theta}|\mathcal{D}) \parallel p(\boldsymbol{\theta}|\mathcal{D})] \triangleq \int q(\boldsymbol{\theta}|\mathcal{D}) \, \log(q(\boldsymbol{\theta}|\mathcal{D})/p(\boldsymbol{\theta}|\mathcal{D})) \, \mathrm{d}\boldsymbol{\theta}$ or, equivalently, maximizing the *evidence lower bound* (ELBO) $\mathcal{L}$ [2]:

$$\mathcal{L} \triangleq \int q(\boldsymbol{\theta}|\mathcal{D}) \, \log p(\mathcal{D}|\boldsymbol{\theta}) \, \mathrm{d}\boldsymbol{\theta} - \mathrm{KL}[q(\boldsymbol{\theta}|\mathcal{D}) \parallel p(\boldsymbol{\theta})] \ . \tag{1}$$

Such an equivalence follows directly from $\mathcal{L} = \log p(\mathcal{D}) - \mathrm{KL}[q(\boldsymbol{\theta}|\mathcal{D}) \parallel p(\boldsymbol{\theta}|\mathcal{D})]$ where the log-marginal likelihood $\log p(\mathcal{D})$ is independent of $q(\boldsymbol{\theta}|\mathcal{D})$. Since $\mathrm{KL}[q(\boldsymbol{\theta}|\mathcal{D}) \parallel p(\boldsymbol{\theta}|\mathcal{D})] \geq 0$, the ELBO $\mathcal{L}$ is a lower bound of $\log p(\mathcal{D})$. The ELBO $\mathcal{L}$ in (1) can be interpreted as a trade-off between attaining a higher likelihood of $\mathcal{D}$ (first term) vs. not entirely forgetting the prior belief $p(\boldsymbol{\theta})$ (second term).

When the ELBO $\mathcal{L}$ (1) cannot be evaluated in closed form, it can be maximized using *stochastic gradient ascent* (SGA) by approximating the expectation in

$$\mathcal{L} = \mathbb{E}_{q(\boldsymbol{\theta}|\mathcal{D})}[\log p(\mathcal{D}|\boldsymbol{\theta}) + \log(p(\boldsymbol{\theta})/q(\boldsymbol{\theta}|\mathcal{D}))] = \int q(\boldsymbol{\theta}|\mathcal{D}) \, (\log p(\mathcal{D}|\boldsymbol{\theta}) + \log(p(\boldsymbol{\theta})/q(\boldsymbol{\theta}|\mathcal{D}))) \, \mathrm{d}\boldsymbol{\theta}$$

with stochastic sampling in each iteration of SGA. The approximate posterior belief $q(\boldsymbol{\theta}|\mathcal{D})$ can be represented by a simple distribution (e.g., in the exponential family) for computational ease or a complex distribution (e.g., using generative neural networks) for expressive power. Note that when the distribution of $q(\boldsymbol{\theta}|\mathcal{D})$ is modeled by a generative neural network whose density cannot be evaluated, the ELBO can be maximized with adversarial training by alternating between estimating the log-density ratio $\log(p(\boldsymbol{\theta})/q(\boldsymbol{\theta}|\mathcal{D}))$ and maximizing the ELBO [33]. On the other hand, when the distribution of $q(\boldsymbol{\theta}|\mathcal{D})$ is modeled by a normalizing flow (e.g., *inverse autoregressive flow* (IAF) [21]) whose density can be computed, the ELBO can be maximized with SGA.

# 3 Bayesian Unlearning

## 3.1 Exact Bayesian Unlearning

Let the (*full*) training data $\mathcal{D}$ be partitioned into a small subset $\mathcal{D}_e$ of data to be *erased* and a (large) set $\mathcal{D}_r$ of *remaining* data, i.e., $\mathcal{D} = \mathcal{D}_r \cup \mathcal{D}_e$ and $\mathcal{D}_r \cap \mathcal{D}_e = \emptyset$. The problem of *exact Bayesian unlearning* involves recovering the exact posterior belief $p(\boldsymbol{\theta}|\mathcal{D}_r)$ of model parameters $\boldsymbol{\theta}$ given remaining data $\mathcal{D}_r$ from that given full data $\mathcal{D}$ (i.e., $p(\boldsymbol{\theta}|\mathcal{D})$ assumed to be available) by directly unlearning from erased data $\mathcal{D}_e$. Note that $p(\boldsymbol{\theta}|\mathcal{D}_r)$ can also be obtained from retraining with remaining data $\mathcal{D}_r$, which is computationally costly, as discussed in Sec. 1. By using Bayes' rule and assuming conditional independence between $\mathcal{D}_r$ and $\mathcal{D}_e$ given $\boldsymbol{\theta}$,

$$p(\boldsymbol{\theta}|\mathcal{D}_r) \;=\; p(\boldsymbol{\theta}|\mathcal{D})\, p(\mathcal{D}_e|\mathcal{D}_r)/p(\mathcal{D}_e|\boldsymbol{\theta}) \;\propto\; p(\boldsymbol{\theta}|\mathcal{D})/p(\mathcal{D}_e|\boldsymbol{\theta}) \;. \tag{2}$$

When the model parameters $\boldsymbol{\theta}$ are discrete-valued, $p(\boldsymbol{\theta}|\mathcal{D}_r)$ can be obtained from (2) directly. The use of a conjugate prior also makes unlearning relatively simple. We will investigate the more interesting case of a non-conjugate prior in the rest of Sec. 3.

## 3.2 Approximate Bayesian Unlearning with Exact Posterior Belief $p(\boldsymbol{\theta}|\mathcal{D})$

The problem of *approximate Bayesian unlearning* differs from that of exact Bayesian unlearning (Sec. 3.1) in that only the approximate posterior belief $q_u(\boldsymbol{\theta}|\mathcal{D}_r)$ (instead of the exact one $p(\boldsymbol{\theta}|\mathcal{D}_r)$) can be recovered by directly unlearning from erased data $\mathcal{D}_e$. Since existing unlearning methods often use their model predictions to construct their loss functions [3, 4, 12, 14], we have initially considered doing likewise (albeit in the Bayesian context) by defining the loss function as the KL divergence between the approximate predictive distribution $q_u(y|\mathcal{D}_r) \triangleq \int p(y|\boldsymbol{\theta})\, q_u(\boldsymbol{\theta}|\mathcal{D}_r)\, \mathrm{d}\boldsymbol{\theta}$ vs. the exact predictive distribution $p(y|\mathcal{D}_r) = \int p(y|\boldsymbol{\theta})\, p(\boldsymbol{\theta}|\mathcal{D}_r)\, \mathrm{d}\boldsymbol{\theta}$ where the observation $y$ (i.e., drawn from a model with parameters $\boldsymbol{\theta}$) is conditionally independent of $\mathcal{D}_r$ given $\boldsymbol{\theta}$. However, it may not be possible to evaluate these predictive distributions in closed form, hence making the optimization of this loss function computationally difficult. Fortunately, such a loss function can be bounded from above by the KL divergence between posterior beliefs $q_u(\boldsymbol{\theta}|\mathcal{D}_r)$ vs. $p(\boldsymbol{\theta}|\mathcal{D}_r)$, as proven in Appendix A:

**Proposition 1.** $\mathrm{KL}[q_u(y|\mathcal{D}_r) \parallel p(y|\mathcal{D}_r)] \leq \mathrm{KL}[q_u(\boldsymbol{\theta}|\mathcal{D}_r) \parallel p(\boldsymbol{\theta}|\mathcal{D}_r)]$ .[2]

Proposition 1 reveals that reducing $\mathrm{KL}[q_u(\boldsymbol{\theta}|\mathcal{D}_r) \parallel p(\boldsymbol{\theta}|\mathcal{D}_r)]$ decreases $\mathrm{KL}[q_u(y|\mathcal{D}_r) \parallel p(y|\mathcal{D}_r)]$, thus motivating its use as the loss function instead. In particular, it follows immediately from our result below (i.e., proven in Appendix B) that minimizing $\mathrm{KL}[q_u(\boldsymbol{\theta}|\mathcal{D}_r) \parallel p(\boldsymbol{\theta}|\mathcal{D}_r)]$ is equivalent to *minimizing* a counterpart of the ELBO called the *evidence upper bound* (EUBO) $\mathcal{U}$:

**Proposition 2.** Define the EUBO $\mathcal{U}$ as

$$\mathcal{U} \triangleq \int q_u(\boldsymbol{\theta}|\mathcal{D}_r)\, \log p(\mathcal{D}_e|\boldsymbol{\theta})\, \mathrm{d}\boldsymbol{\theta} + \mathrm{KL}[q_u(\boldsymbol{\theta}|\mathcal{D}_r) \parallel p(\boldsymbol{\theta}|\mathcal{D})] \;. \tag{3}$$

Then, $\mathcal{U} = \log p(\mathcal{D}_e|\mathcal{D}_r) + \mathrm{KL}[q_u(\boldsymbol{\theta}|\mathcal{D}_r) \parallel p(\boldsymbol{\theta}|\mathcal{D}_r)] \geq \log p(\mathcal{D}_e|\mathcal{D}_r)$ such that $p(\mathcal{D}_e|\mathcal{D}_r)$ is independent of $q_u(\boldsymbol{\theta}|\mathcal{D}_r)$.

From Proposition 2, minimizing EUBO (3) is equivalent to minimizing $\mathrm{KL}[q_u(\boldsymbol{\theta}|\mathcal{D}_r) \parallel p(\boldsymbol{\theta}|\mathcal{D}_r)]$ which is precisely achieved using VI (i.e., by maximizing ELBO (1)) from retraining with remaining data $\mathcal{D}_r$. This is illustrated in Fig. 1a where unlearning from $\mathcal{D}_e$ by minimizing EUBO maximizes ELBO w.r.t. $\mathcal{D}_r$; in Fig. 1b, retraining with $\mathcal{D}_r$ by maximizing ELBO minimizes EUBO w.r.t. $\mathcal{D}_e$.

The EUBO $\mathcal{U}$ (3) can be interpreted as a trade-off between fully unlearning from erased data $\mathcal{D}_e$ (first term) vs. not entirely forgetting the exact posterior belief $p(\boldsymbol{\theta}|\mathcal{D})$ given the full data $\mathcal{D}$ (i.e., including the remaining data $\mathcal{D}_r$) (second term). The latter can be viewed as a regularization term to prevent catastrophic unlearning[1] (i.e., potentially induced by the former) that *naturally* results from minimizing our loss function $\mathrm{KL}[q_u(\boldsymbol{\theta}|\mathcal{D}_r) \parallel p(\boldsymbol{\theta}|\mathcal{D}_r)]$, which differs from the works of [9, 13] needing to "patch up" their loss functions (Sec. 1). Generative models can be used to model the approximate posterior belief $q_u(\boldsymbol{\theta}|\mathcal{D}_r)$ in the EUBO $\mathcal{U}$ (3) in the same way as that in the ELBO $\mathcal{L}$ (1).

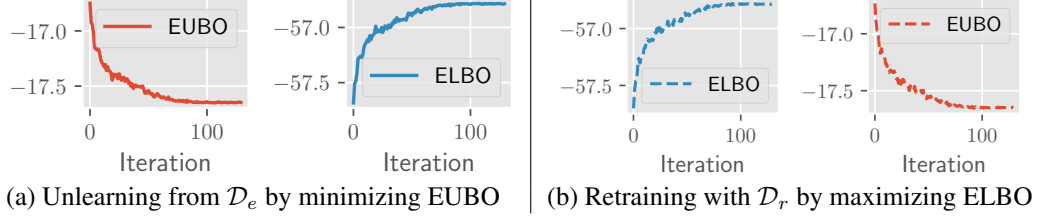

(a) Unlearning from $\mathcal{D}_e$ by minimizing EUBO     (b) Retraining with $\mathcal{D}_r$ by maximizing ELBO

Figure 1: Plots of EUBO and ELBO when (a) unlearning from $\mathcal{D}_e$ and (b) retraining with $\mathcal{D}_r$.

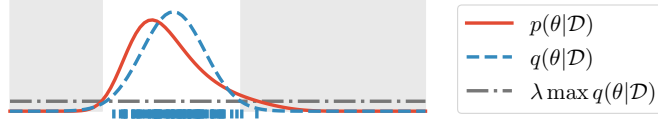

Figure 2: Plot of $q(\boldsymbol{\theta}|\mathcal{D})$ learned using VI. Gray shaded region corresponds to values of $\boldsymbol{\theta}$ where $q(\boldsymbol{\theta}|\mathcal{D}) \leq \lambda \max_{\boldsymbol{\theta}'} q(\boldsymbol{\theta}'|\mathcal{D})$. Vertical blue strips on horizontal axis show 100 samples of $\boldsymbol{\theta} \sim q(\boldsymbol{\theta}|\mathcal{D})$.

### 3.3 Approximate Bayesian Unlearning with Approximate Posterior Belief $q(\boldsymbol{\theta}|\mathcal{D})$

Often, in model training, only an approximate posterior belief[3] $q(\boldsymbol{\theta}|\mathcal{D})$ (instead of the exact $p(\boldsymbol{\theta}|\mathcal{D})$ in Sec. 3.2) of model parameters $\boldsymbol{\theta}$ given full data $\mathcal{D}$ can be learned, say, using VI by maximizing the ELBO (Sec. 2). Our proposed unlearning methods are parsimonious in requiring only $q(\boldsymbol{\theta}|\mathcal{D})$ and erased data $\mathcal{D}_e$ to be available, which makes unlearning even more challenging since there is no further information about $p(\boldsymbol{\theta}|\mathcal{D})$ nor the difference between $p(\boldsymbol{\theta}|\mathcal{D})$ vs. $q(\boldsymbol{\theta}|\mathcal{D})$. So, we estimate the unknown $p(\boldsymbol{\theta}|\mathcal{D}_r)$ (2) with

$$\tilde{p}(\boldsymbol{\theta}|\mathcal{D}_r) \quad \propto \quad q(\boldsymbol{\theta}|\mathcal{D})/p(\mathcal{D}_e|\boldsymbol{\theta}) \tag{4}$$

and minimize the KL divergence between the approximate posterior belief recovered by directly unlearning from erased data $\mathcal{D}_e$ vs. $\tilde{p}(\boldsymbol{\theta}|\mathcal{D}_r)$ (4) instead. We will discuss two novel tricks below to alleviate the undesirable consequence of using $\tilde{p}(\boldsymbol{\theta}|\mathcal{D}_r)$ instead of the unknown $p(\boldsymbol{\theta}|\mathcal{D}_r)$ (2).

#### 3.3.1 EUBO with Adjusted Likelihood

Let the loss function $\mathrm{KL}[\tilde{q}_u(\boldsymbol{\theta}|\mathcal{D}_r) \| \tilde{p}(\boldsymbol{\theta}|\mathcal{D}_r)]$ be minimized w.r.t. the approximate posterior belief $\tilde{q}_u(\boldsymbol{\theta}|\mathcal{D}_r)$ that is recovered by directly unlearning from erased data $\mathcal{D}_e$. Similar to Proposition 2, $\tilde{q}_u(\boldsymbol{\theta}|\mathcal{D}_r)$ can be optimized by minimizing the following EUBO:

$$\widetilde{\mathcal{U}} \triangleq \int \tilde{q}_u(\boldsymbol{\theta}|\mathcal{D}_r) \, \log p(\mathcal{D}_e|\boldsymbol{\theta}) \, \mathrm{d}\boldsymbol{\theta} + \mathrm{KL}[\tilde{q}_u(\boldsymbol{\theta}|\mathcal{D}_r) \| q(\boldsymbol{\theta}|\mathcal{D})] \tag{5}$$

which follows from simply replacing the unknown $p(\boldsymbol{\theta}|\mathcal{D})$ in $\mathcal{U}$ (3) with $q(\boldsymbol{\theta}|\mathcal{D})$. We discuss the difference between $p(\boldsymbol{\theta}|\mathcal{D})$ vs. $q(\boldsymbol{\theta}|\mathcal{D})$ in the remark below:

**Remark 1.** We analyze two possible sources of inaccuracy in $q(\boldsymbol{\theta}|\mathcal{D})$ that is learned using VI by minimizing the loss function $\mathrm{KL}[q(\boldsymbol{\theta}|\mathcal{D}) \| p(\boldsymbol{\theta}|\mathcal{D})]$ (Sec. 2). Firstly, $q(\boldsymbol{\theta}|\mathcal{D})$ often underestimates the variance of $p(\boldsymbol{\theta}|\mathcal{D})$: Though $q(\boldsymbol{\theta}|\mathcal{D})$ tends to be close to 0 at values of $\boldsymbol{\theta}$ where $p(\boldsymbol{\theta}|\mathcal{D})$ is close to 0, the reverse is not enforced [1] (see, for example, Fig. 2). So, $q(\boldsymbol{\theta}|\mathcal{D})$ can differ from $p(\boldsymbol{\theta}|\mathcal{D})$ at values of $\boldsymbol{\theta}$ where $q(\boldsymbol{\theta}|\mathcal{D})$ is close to 0. Secondly, if $q(\boldsymbol{\theta}|\mathcal{D})$ is learned through stochastic optimization of the ELBO (i.e., with stochastic samples of $\boldsymbol{\theta} \sim q(\boldsymbol{\theta}|\mathcal{D})$ in each iteration of SGA), then it is unlikely that the ELBO is maximized using samples of $\boldsymbol{\theta}$ with small $q(\boldsymbol{\theta}|\mathcal{D})$ (Fig. 2). Thus, both sources of inaccuracy primarily occur at values of $\boldsymbol{\theta}$ with small $q(\boldsymbol{\theta}|\mathcal{D})$. Though it can also be inaccurate at values of $\boldsymbol{\theta}$ with large $q(\boldsymbol{\theta}|\mathcal{D})$, such an inaccuracy can be reduced by representing $q(\boldsymbol{\theta}|\mathcal{D})$ with a complex distribution (Sec. 2).

Remark 1 motivates us to curb unlearning at values of $\boldsymbol{\theta}$ with small $q(\boldsymbol{\theta}|\mathcal{D})$ by proposing our first novel trick of an adjusted likelihood of the erased data:

$$p_{\mathrm{adj}}(\mathcal{D}_e|\boldsymbol{\theta}; \lambda) \triangleq \begin{cases} p(\mathcal{D}_e|\boldsymbol{\theta}) & \text{if } q(\boldsymbol{\theta}|\mathcal{D}) > \lambda \max_{\boldsymbol{\theta}'} q(\boldsymbol{\theta}'|\mathcal{D}) \,, \\ 1 & \text{otherwise (i.e., shaded area in Fig. 2) ;} \end{cases} \tag{6}$$

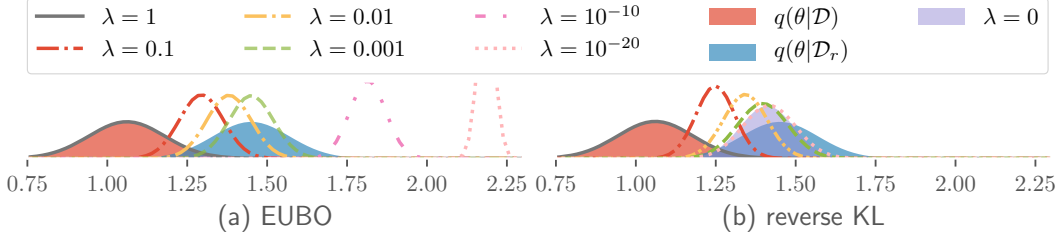

Figure 3: Plot of approximate posterior beliefs with varying $\lambda$ obtained by minimizing (a) EUBO (i.e., $\tilde{q}_u(\boldsymbol{\theta}|\mathcal{D}_r; \lambda)$) and (b) reverse KL (i.e., $\tilde{q}_v(\boldsymbol{\theta}|\mathcal{D}_r; \lambda)$); horizontal axis denotes $\theta = \alpha$. In (a), a huge probability mass of $\tilde{q}_u(\boldsymbol{\theta}|\mathcal{D}_r, \lambda = 0)$ is at large values of $\alpha$ beyond the plotting area and the top of the plot of $\tilde{q}_u(\boldsymbol{\theta}|\mathcal{D}_r, \lambda = 10^{-20})$ is cut off due to lack of space.

for any $\boldsymbol{\theta}$ where $\lambda \in [0, 1]$ controls the magnitude of a threshold under which $q(\boldsymbol{\theta}|\mathcal{D})$ is considered small. To understand the effect of $\lambda$, let $\tilde{p}_{\text{adj}}(\boldsymbol{\theta}|\mathcal{D}_r; \lambda) \propto q(\boldsymbol{\theta}|\mathcal{D})/p_{\text{adj}}(\mathcal{D}_e|\boldsymbol{\theta}; \lambda)$, i.e., by replacing $p(\mathcal{D}_e|\boldsymbol{\theta})$ in (4) with $p_{\text{adj}}(\mathcal{D}_e|\boldsymbol{\theta}; \lambda)$. Then, using (6),

$$\tilde{p}_{\text{adj}}(\boldsymbol{\theta}|\mathcal{D}_r; \lambda) \propto \begin{cases} q(\boldsymbol{\theta}|\mathcal{D})/p(\mathcal{D}_e|\boldsymbol{\theta}) & \text{if } q(\boldsymbol{\theta}|\mathcal{D}) > \lambda \max_{\boldsymbol{\theta}'} q(\boldsymbol{\theta}'|\mathcal{D}) , \\ q(\boldsymbol{\theta}|\mathcal{D}) & \text{otherwise (i.e., shaded area in Fig. 2) .} \end{cases} \quad (7)$$

According to (7), unlearning is therefore focused at values of $\boldsymbol{\theta}$ with sufficiently large $q(\boldsymbol{\theta}|\mathcal{D})$ (i.e., $q(\boldsymbol{\theta}|\mathcal{D}) > \lambda \max_{\boldsymbol{\theta}'} q(\boldsymbol{\theta}'|\mathcal{D})$). Let the loss function $\text{KL}[\tilde{q}_u(\boldsymbol{\theta}|\mathcal{D}_r; \lambda) \| \tilde{p}_{\text{adj}}(\boldsymbol{\theta}|\mathcal{D}_r; \lambda)]$ be minimized w.r.t. the approximate posterior belief $\tilde{q}_u(\boldsymbol{\theta}|\mathcal{D}_r; \lambda)$ that is recovered by directly unlearning from erased data $\mathcal{D}_e$. Similar to (5), $\tilde{q}_u(\boldsymbol{\theta}|\mathcal{D}_r; \lambda)$ can be optimized by minimizing the following EUBO:

$$\widetilde{\mathcal{U}}_{\text{adj}}(\lambda) \triangleq \int \tilde{q}_u(\boldsymbol{\theta}|\mathcal{D}_r; \lambda) \, \log p_{\text{adj}}(\mathcal{D}_e|\boldsymbol{\theta}; \lambda) \, \mathrm{d}\boldsymbol{\theta} + \text{KL}[\tilde{q}_u(\boldsymbol{\theta}|\mathcal{D}_r; \lambda) \| q(\boldsymbol{\theta}|\mathcal{D})] \quad (8)$$

which follows from replacing $p(\mathcal{D}_e|\boldsymbol{\theta})$ in (5) with $p_{\text{adj}}(\mathcal{D}_e|\boldsymbol{\theta}; \lambda)$. Note that $\tilde{q}_u(\boldsymbol{\theta}|\mathcal{D}_r; \lambda)$ can be represented by a simple distribution (e.g., Gaussian) or a complex one (e.g., generative neural network, IAF). We initialize $\tilde{q}_u(\boldsymbol{\theta}|\mathcal{D}_r; \lambda)$ at $q(\boldsymbol{\theta}|\mathcal{D})$ for achieving empirically faster convergence. When $\lambda = 0, \widetilde{\mathcal{U}}_{\text{adj}}(\lambda = 0)$ reduces to $\widetilde{\mathcal{U}}$ (5), i.e., EUBO is minimized without the adjusted likelihood. As a result, $\tilde{q}_u(\boldsymbol{\theta}|\mathcal{D}_r; \lambda = 0) = \tilde{q}_u(\boldsymbol{\theta}|\mathcal{D}_r)$. As $\lambda$ increases, unlearning is focused on a smaller and smaller region of $\boldsymbol{\theta}$ with sufficiently large $q(\boldsymbol{\theta}|\mathcal{D})$. When $\lambda$ reaches 1, no unlearning is performed since $\tilde{p}_{\text{adj}}(\boldsymbol{\theta}|\mathcal{D}_r; \lambda = 1) = q(\boldsymbol{\theta}|\mathcal{D})$, which results in $\tilde{q}_u(\boldsymbol{\theta}|\mathcal{D}_r; \lambda = 1) = q(\boldsymbol{\theta}|\mathcal{D})$ minimizing the loss function $\text{KL}[\tilde{q}_u(\boldsymbol{\theta}|\mathcal{D}_r; \lambda = 1) \| \tilde{p}_{\text{adj}}(\boldsymbol{\theta}|\mathcal{D}_r; \lambda = 1)]$.

**Example 1.** To visualize the effect of varying $\lambda$ on $\tilde{q}_u(\boldsymbol{\theta}|\mathcal{D}_r; \lambda)$, we consider learning the shape $\alpha$ of a Gamma distribution with a known rate (i.e., $\boldsymbol{\theta} = \alpha$): $\mathcal{D}$ are 20 samples of the Gamma distribution, $\mathcal{D}_e$ are the smallest 5 samples in $\mathcal{D}$, and the (non-conjugate) prior belief and approximate posterior beliefs of $\alpha$ are all Gaussians. Fig. 3a shows the approximate posterior beliefs $\tilde{q}_u(\boldsymbol{\theta}|\mathcal{D}_r; \lambda)$ with varying $\lambda$ as well as $q(\boldsymbol{\theta}|\mathcal{D})$ and $q(\boldsymbol{\theta}|\mathcal{D}_r)$ learned using VI. As explained above, $\tilde{q}_u(\boldsymbol{\theta}|\mathcal{D}_r, \lambda = 1) = q(\boldsymbol{\theta}|\mathcal{D})$. When $\lambda = 0.001$, $\tilde{q}_u(\boldsymbol{\theta}|\mathcal{D}_r, \lambda = 0.001)$ is close to $q(\boldsymbol{\theta}|\mathcal{D}_r)$. However, as $\lambda$ decreases to 0, $\tilde{q}_u(\boldsymbol{\theta}|\mathcal{D}_r, \lambda)$ moves away from $q(\boldsymbol{\theta}|\mathcal{D}_r)$.

The optimized $\tilde{q}_u(\boldsymbol{\theta}|\mathcal{D}_r; \lambda)$ suffers from the same issue of underestimating the variance as $q(\boldsymbol{\theta}|\mathcal{D})$ learned using VI (see Remark 1), especially when $\lambda$ tends to 0 (e.g., see $\tilde{q}_u(\boldsymbol{\theta}|\mathcal{D}_r; \lambda = 10^{-20})$ in Fig. 3a). Though this issue can be mitigated by tuning $\lambda$ in the adjusted likelihood (6), we may not want to risk facing the consequence of picking a value of $\lambda$ that is too small. So, in Sec. 3.3.2, we will propose another novel trick that is not inconvenienced by this issue.

### 3.3.2 Reverse KL

Let the loss function be the reverse KL divergence $\text{KL}[\tilde{p}(\boldsymbol{\theta}|\mathcal{D}_r) \| \tilde{q}_v(\boldsymbol{\theta}|\mathcal{D}_r)]$ that is minimized w.r.t. the approximate posterior belief $\tilde{q}_v(\boldsymbol{\theta}|\mathcal{D}_r)$ recovered by directly unlearning from erased data $\mathcal{D}_e$. In contrast to the optimized $\tilde{q}_u(\boldsymbol{\theta}|\mathcal{D}_r; \lambda)$ from minimizing EUBO (8), the optimized $\tilde{q}_v(\boldsymbol{\theta}|\mathcal{D}_r)$ from minimizing the reverse KL divergence overestimates (instead of underestimates) the variance of $\tilde{p}(\boldsymbol{\theta}|\mathcal{D}_r)$ [1]: If $\tilde{p}(\boldsymbol{\theta}|\mathcal{D}_r)$ is close to 0, then $\tilde{q}_v(\boldsymbol{\theta}|\mathcal{D}_r)$ is not necessarily close to 0. From (4), the reverse KL divergence can be expressed as follows:

$$\text{KL}[\tilde{p}(\boldsymbol{\theta}|\mathcal{D}_r) \| \tilde{q}_v(\boldsymbol{\theta}|\mathcal{D}_r)] = C_0 - C_1 \, \mathbb{E}_{q(\boldsymbol{\theta}|\mathcal{D})} [(\log \tilde{q}_v(\boldsymbol{\theta}|\mathcal{D}_r))/p(\mathcal{D}_e|\boldsymbol{\theta})] \quad (9)$$

where $C_0$ and $C_1$ are constants independent of $\tilde{q}_v(\boldsymbol{\theta}|\mathcal{D}_r)$. So, the reverse KL divergence (9) can be minimized by maximizing $\mathbb{E}_{q(\boldsymbol{\theta}|\mathcal{D})}[(\log \tilde{q}_v(\boldsymbol{\theta}|\mathcal{D}_r))/p(\mathcal{D}_e|\boldsymbol{\theta})]$ with *stochastic gradient ascent* (SGA). We also initialize $\tilde{q}_v(\boldsymbol{\theta}|\mathcal{D}_r)$ at $q(\boldsymbol{\theta}|\mathcal{D})$ for achieving empirically faster convergence. Since stochastic optimization is performed with samples of $\boldsymbol{\theta} \sim q(\boldsymbol{\theta}|\mathcal{D})$ in each iteration of SGA, it is unlikely that the reverse KL divergence (9) is minimized using samples of $\boldsymbol{\theta}$ with small $q(\boldsymbol{\theta}|\mathcal{D})$. This naturally curbs unlearning at values of $\boldsymbol{\theta}$ with small $q(\boldsymbol{\theta}|\mathcal{D})$, as motivated by Remark 1. On the other hand, it is still possible to employ the same trick of adjusted likelihood (Sec. 3.3.1) by minimizing the reverse KL divergence $\mathrm{KL}[\tilde{p}_{\mathrm{adj}}(\boldsymbol{\theta}|\mathcal{D}_r; \lambda) \,\|\, \tilde{q}_v(\boldsymbol{\theta}|\mathcal{D}_r; \lambda)]$ as the loss function or, equivalently, maximizing $\mathbb{E}_{q(\boldsymbol{\theta}|\mathcal{D})}[(\log \tilde{q}_v(\boldsymbol{\theta}|\mathcal{D}_r; \lambda))/p_{\mathrm{adj}}(\mathcal{D}_e|\boldsymbol{\theta}; \lambda)]$ where $p_{\mathrm{adj}}(\mathcal{D}_e|\boldsymbol{\theta}; \lambda)$ and $\tilde{p}_{\mathrm{adj}}(\boldsymbol{\theta}|\mathcal{D}_r; \lambda)$ are previously defined in (6) and (7), respectively. The role of $\lambda$ here is the same as that in (8).

To illustrate the difference between minimizing the reverse KL divergence (9) and EUBO (8), Fig. 3b shows the approximate posterior beliefs $\tilde{q}_v(\boldsymbol{\theta}|\mathcal{D}_r; \lambda)$ with varying $\lambda$. It can be observed that $\tilde{q}_v(\boldsymbol{\theta}|\mathcal{D}_r; \lambda = 1) = q(\boldsymbol{\theta}|\mathcal{D})$ (i.e., no unlearning). In contrast to minimizing EUBO (Fig. 3a), as $\lambda$ decreases to 0, $\tilde{q}_v(\boldsymbol{\theta}|\mathcal{D}_r; \lambda)$ does not deviate that much from $q(\boldsymbol{\theta}|\mathcal{D}_r)$, even when $\lambda = 0$ (i.e., the reverse KL divergence is minimized without the adjusted likelihood). This is because the optimized $\tilde{q}_v(\boldsymbol{\theta}|\mathcal{D}_r; \lambda)$ is naturally more protected from both sources of inaccuracy (Remark 1), as explained above. Hence, we do not have to be as concerned about picking a small value of $\lambda$, which is also consistently observed in our experiments (Sec. 4).

## 4    Experiments and Discussion

This section empirically demonstrates our unlearning methods on Bayesian models such as sparse Gaussian process and logistic regression using synthetic and real-world datasets. Further experimental results on Bayesian linear regression and with a bimodal posterior belief are reported in Appendices C and D, respectively. In our experiments, each dataset comprises pairs of input $\mathbf{x}$ and its corresponding output/observation $y_{\mathbf{x}}$. We use RMSProp as the SGA algorithm with a learning rate of $10^{-4}$. To assess the performance of our unlearning methods (i.e., by directly unlearning from erased data $\mathcal{D}_e$), we consider the difference between their induced predictive distributions vs. that obtained using VI from retraining with remaining data $\mathcal{D}_r$, as motivated from Sec. 3.2. To do this, we use a **performance metric** that measures the KL divergence between the approximate predictive distributions

$$\tilde{q}_u(y_{\mathbf{x}}|\mathcal{D}_r) \triangleq \int p(y_{\mathbf{x}}|\boldsymbol{\theta})\, \tilde{q}_u(\boldsymbol{\theta}|\mathcal{D}_r; \lambda)\, \mathrm{d}\boldsymbol{\theta} \quad \text{or} \quad \tilde{q}_v(y_{\mathbf{x}}|\mathcal{D}_r) \triangleq \int p(y_{\mathbf{x}}|\boldsymbol{\theta})\, \tilde{q}_v(\boldsymbol{\theta}|\mathcal{D}_r; \lambda)\, \mathrm{d}\boldsymbol{\theta}$$

vs. $q(y_{\mathbf{x}}|\mathcal{D}_r) \triangleq \int p(y_{\mathbf{x}}|\boldsymbol{\theta})\, q(\boldsymbol{\theta}|\mathcal{D}_r)\, \mathrm{d}\boldsymbol{\theta}$ where $\tilde{q}_u(\boldsymbol{\theta}|\mathcal{D}_r; \lambda)$ and $\tilde{q}_v(\boldsymbol{\theta}|\mathcal{D}_r; \lambda)$ are optimized by, respectively, minimizing EUBO (8) and *reverse KL* (rKL) divergence (9) while requiring only $q(\boldsymbol{\theta}|\mathcal{D})$ and erased data $\mathcal{D}_e$ (Sec. 3.3), and $q(\boldsymbol{\theta}|\mathcal{D}_r)$ is learned using VI (Sec. 2). The above predictive distributions are computed via sampling with 100 samples of $\boldsymbol{\theta}$. For tractability reasons, we evaluate the above performance metric over finite input domains, specifically, over that in $\mathcal{D}_e$ and $\mathcal{D}_r$, which allows us to assess the performance of our unlearning methods on both the erased and remaining data, i.e., whether they can fully unlearn from $\mathcal{D}_e$ yet not forget nor catastrophically unlearn from $\mathcal{D}_r$, respectively. For example, the performance of our EUBO-based unlearning method over $\mathcal{D}_e$ is shown as an average (with standard deviation) of the KL divergences between $\tilde{q}_u(y_{\mathbf{x}}|\mathcal{D}_r)$ vs. $q(y_{\mathbf{x}}|\mathcal{D}_r)$ over all $(\mathbf{x}, y_{\mathbf{x}}) \in \mathcal{D}_e$. We also plot an average (with standard deviation) of the KL divergences between $q(y_{\mathbf{x}}|\mathcal{D})$ vs. $q(y_{\mathbf{x}}|\mathcal{D}_r)$ over $\mathcal{D}_r$ and $\mathcal{D}_e$ as *baselines* (i.e., representing no unlearning), which is expected to be larger than that of our unlearning methods (i.e., if performing well) and labeled as *full* in the plots below.

### 4.1    Sparse Gaussian Process (GP) Classification with Synthetic Moon Dataset

This experiment is about unlearning a binary classifier that is previously trained with the synthetic moon dataset (Fig. 4a). The probability of input $\mathbf{x} \in \mathbb{R}^2$ being in the 'blue' class (i.e., $y_{\mathbf{x}} = 1$ and denoted by blue dots in Fig. 4a) is defined as $1/(1 + \exp(f_{\mathbf{x}}))$ where $f_{\mathbf{x}}$ is a latent function modeled by a sparse GP [27], which is elaborated in Appendix E. The parameters $\boldsymbol{\theta}$ of the sparse GP consist of 20 inducing variables; the approximate posterior beliefs of $\boldsymbol{\theta}$ are thus multivariate Gaussians (with full covariance matrices), as shown in Appendix E. By comparing Figs. 4b and 4c, it can be observed that after erasing $\mathcal{D}_e$ (i.e., mainly in 'yellow' class), $q(y_{\mathbf{x}} = 1|\mathcal{D}_r)$ increases at $\mathbf{x} \in \mathcal{D}_e$. Figs. 4d and 4e show results of the performance of our EUBO- and rKL-based unlearning methods

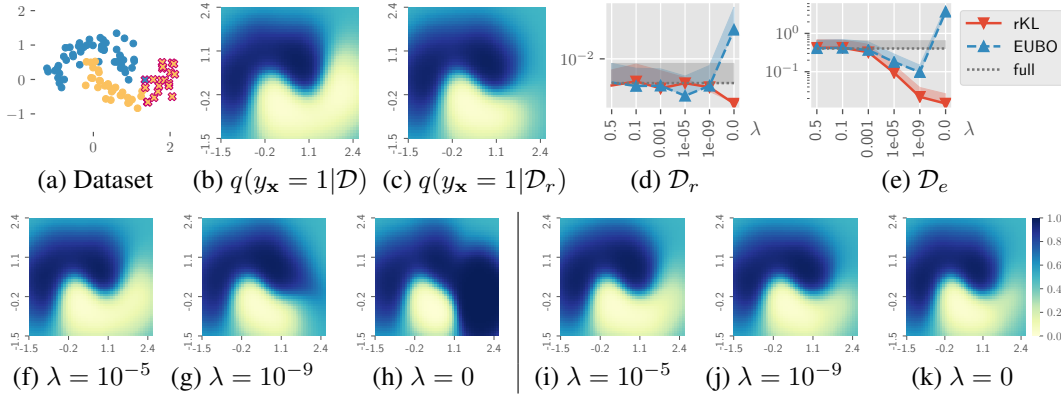

(a) Dataset    (b) $q(y_{\mathbf{x}} = 1|\mathcal{D})$    (c) $q(y_{\mathbf{x}} = 1|\mathcal{D}_r)$    (d) $\mathcal{D}_r$    (e) $\mathcal{D}_e$

(f) $\lambda = 10^{-5}$    (g) $\lambda = 10^{-9}$    (h) $\lambda = 0$    (i) $\lambda = 10^{-5}$    (j) $\lambda = 10^{-9}$    (k) $\lambda = 0$

Figure 4: Plots of (a) synthetic moon dataset with erased data $\mathcal{D}_e$ (crosses) and remaining data $\mathcal{D}_r$ (dots), and of predictive distributions obtained using VI from (b) training with full data $\mathcal{D}$ and (c) retraining with $\mathcal{D}_r$. Graphs of averaged KL divergence vs. $\lambda$ achieved by EUBO, rKL, and $q(\boldsymbol{\theta}|\mathcal{D})$ (i.e., baseline labeled as *full*) over (d) $\mathcal{D}_r$ and (e) $\mathcal{D}_e$. Plots of predictive distributions (f-h) $\tilde{q}_u(y_{\mathbf{x}} = 1|\mathcal{D}_r)$ and (i-k) $\tilde{q}_v(y_{\mathbf{x}} = 1|\mathcal{D}_r)$ induced, respectively, by EUBO and rKL for varying $\lambda$.

over $\mathcal{D}_r$ and $\mathcal{D}_e$ with varying $\lambda$, respectively.[4] When $\lambda = 10^{-9}$, EUBO performs reasonably well (compare Figs. 4g vs. 4c) as its averaged KL divergence is smaller than that of $q(\boldsymbol{\theta}|\mathcal{D})$ (i.e., baseline labeled as *full*). When $\lambda = 0$, EUBO performs poorly (compare Figs. 4h vs. 4c) as its averaged KL divergence is much larger than that of $q(\boldsymbol{\theta}|\mathcal{D})$, as shown in Figs. 4d and 4e. This agrees with our discussion of the issue with picking too small a value of $\lambda$ for EUBO at the end of Sec. 3.3.1. In particular, catastrophic unlearning is observed as the input region containing $\mathcal{D}_e$ (i.e., mainly in 'yellow' class) has a high probability in 'blue' class after unlearning in Fig. 4h. On the other hand, when $\lambda = 0$, rKL performs well (compare Figs. 4k vs. 4c) as its KL divergence is much smaller than that of $q(\boldsymbol{\theta}|\mathcal{D})$, as seen in Figs. 4d and 4e. This agrees with our discussion at the end of Sec. 3.3.2 that rKL can work well without needing the adjusted likelihood.

One may question how the performance of our unlearning methods would vary when erasing a large quantity of data or with different distributions of erased data (e.g., erasing the data randomly vs. deliberately erasing all data in a given class). To address this question, we have discovered that a key factor influencing their unlearning performance in these scenarios is the difference between the posterior beliefs of model parameters $\boldsymbol{\theta}$ given remaining data $\mathcal{D}_r$ vs. that given full data $\mathcal{D}$, especially at values of $\boldsymbol{\theta}$ with small $q(\boldsymbol{\theta}|\mathcal{D})$ since unlearning in such a region is curbed by the adjusted likelihood and reverse KL. In practice, we expect such a difference not to be large due to the small quantity of erased data and redundancy in real-world datasets. We will present the details of this study in Appendix F due to lack of space by considering how much $\mathcal{D}_e$ reduces the entropy of $\boldsymbol{\theta}$ given $\mathcal{D}_r$.

## 4.2 Logistic Regression with Banknote Authentication Dataset

The banknote authentication dataset [10] of size $|\mathcal{D}| = 1372$ is partitioned into erased data of size $|\mathcal{D}_e| = 412$ and remaining data of size $|\mathcal{D}_r| = 960$. Each input $\mathbf{x}$ comprises 4 features extracted from an image of a banknote and its corresponding binary label $y_{\mathbf{x}}$ indicates whether the banknote is genuine or forged. We use a logistic regression model with 5 parameters that is trained with this dataset. The prior beliefs of the model parameters are independent Gaussians $\mathcal{N}(0, 100)$.

Unlike the previous experiment, the erased data $\mathcal{D}_e$ here is randomly selected and hence does not reduce the entropy of model parameters $\boldsymbol{\theta}$ given $\mathcal{D}_r$ much, as explained in Appendix F; a discussion on erasing informative data (such as that in Sec. 4.1) is in Appendix F. As a result, Figs. 5a and 5b show a very small averaged KL divergence of about $10^{-3}$ between $q(y_{\mathbf{x}}|\mathcal{D})$ vs. $q(y_{\mathbf{x}}|\mathcal{D}_r)$ (i.e., baselines) over $\mathcal{D}_r$ and $\mathcal{D}_e$.[4] Figs. 5a and 5b also show that our unlearning methods do not perform well when using multivariate Gaussians to model the approximate posterior beliefs of $\boldsymbol{\theta}$: While rKL still gives a useful $\tilde{q}_v(\boldsymbol{\theta}|\mathcal{D}_r; \lambda)$ achieving an averaged KL divergence close to that of $q(\boldsymbol{\theta}|\mathcal{D})$, EUBO gives a useless $\tilde{q}_u(\boldsymbol{\theta}|\mathcal{D}_r; \lambda)$ incurring a large averaged KL divergence when $\lambda$ is small. On the other hand,

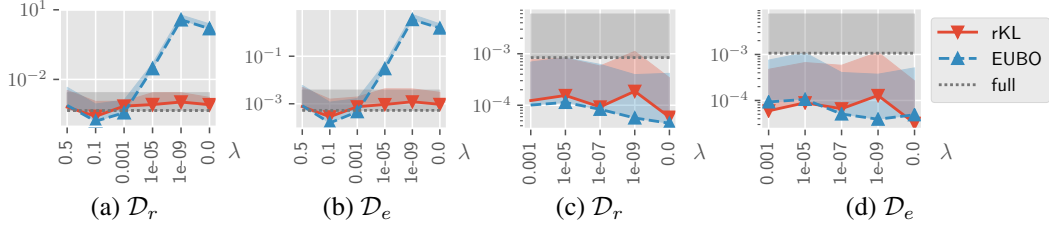

Figure 5: Graphs of averaged KL divergence vs. $\lambda$ achieved by EUBO, rKL, and $q(\boldsymbol{\theta}|\mathcal{D})$ (i.e., baseline labeled as *full*) over $\mathcal{D}_r$ and $\mathcal{D}_e$ for the banknote authentication dataset with the approximate posterior beliefs of model parameters represented by (a-b) multivariate Gaussians and (c-d) normalizing flows.

when more complex models like normalizing flows with the MADE architecture [26] are used to represent the approximate posterior beliefs, EUBO and rKL can unlearn well (Figs. 5c and 5d).

### 4.3 Logistic Regression with Fashion MNIST Dataset

The fashion MNIST dataset of size $|\mathcal{D}| = 60000$ ($28 \times 28$ images of fashion items in 10 classes) is partitioned into erased data of size $|\mathcal{D}_e| = 10000$ and remaining data of size $|\mathcal{D}_r| = 50000$. The classification model is a neural network with 3 fully-connected hidden layers of 128, 128, 64 hidden neurons and a softmax layer to output the 10-class probabilities. The model can be interpreted as one of logistic regression on 64 features generated from the hidden layer of 64 neurons. Since modeling all weights of the neural network as random variables can be costly, we model only 650 weights in the transformation of the 64 features to the inputs of the softmax layer. The other weights remain constant during unlearning and retraining. The prior beliefs of the network weights are $\mathcal{N}(0, 10)$. The approximate posterior beliefs are modeled with independent Gaussians. Though a large part of the network is fixed and we use simple models to represent the approximate posterior beliefs, we show that unlearning is still fairly effective.

As discussed in Sec. 4.1, 4.2, and Appendix F, the random selection of erased data $\mathcal{D}_e$ and redundancy in $\mathcal{D}$ lead to a small averaged KL divergence of about 0.1 between $q(y_\mathbf{x}|\mathcal{D})$ vs. $q(y_\mathbf{x}|\mathcal{D}_r)$ (i.e., baselines) over $\mathcal{D}_r$ and $\mathcal{D}_e$ (Figs. 6a and 6b) despite choosing a relatively large $|\mathcal{D}_e|$. Figs. 6a and 6b show that when $\lambda \geq 10^{-9}$, EUBO and rKL achieve averaged KL divergences comparable to that of $q(\boldsymbol{\theta}|\mathcal{D})$ (i.e., baseline labeled as *full*), hence making their unlearning insignificant.[4] However, at $\lambda = 0$, the unlearning performance of rKL improves by achieving a smaller averaged KL divergence than that of $q(\boldsymbol{\theta}|\mathcal{D})$, while EUBO's performance deteriorates. Their performance can be further improved by using more complex models to represent their approximate posterior beliefs like that in Sec. 4.2, albeit high-dimensional. Figs. 6c and 6d show the class probabilities for two images evaluated at the mean of the approximate posterior beliefs with $\lambda = 0$. We observe that rKL induces the highest class probability for the same class as that of $q(\boldsymbol{\theta}|\mathcal{D}_r)$. The class probabilities for other images are shown in Appendix G. The two images are taken from a separate set of 10000 test images (i.e., different from $\mathcal{D}$) where rKL with $\lambda = 0$ yields the same predictions as $q(\boldsymbol{\theta}|\mathcal{D}_r)$ and $q(\boldsymbol{\theta}|\mathcal{D})$ in, respectively, 99.34% and 99.22% of the test images, the latter of which are contained in the former.

### 4.4 Sparse Gaussian Process (GP) Regression with Airline Dataset

This section illustrates the scalability of unlearning to the massive airline dataset of $\sim 2$ million flights [15]. Training a sparse GP model with this massive dataset is made possible through stochastic VI [15]. Let $\mathcal{X}_\mathbf{u}$ denote the set of 50 inducing inputs in the sparse GP model and $\mathbf{f}_{\mathcal{X}_\mathbf{u}}$ be a vector of corresponding latent function values (i.e., inducing variables). The posterior belief $p(\mathbf{f}_\mathcal{D}, \mathbf{f}_{\mathcal{X}_\mathbf{u}}|\mathcal{D})$ is approximated as $q(\mathbf{f}_\mathcal{D}, \mathbf{f}_{\mathcal{X}_\mathbf{u}}|\mathcal{D}) \triangleq q(\mathbf{f}_{\mathcal{X}_\mathbf{u}}|\mathcal{D}) \, p(\mathbf{f}_\mathcal{D}|\mathbf{f}_{\mathcal{X}_\mathbf{u}})$ where $\mathbf{f}_\mathcal{D} \triangleq (f_\mathbf{x})_{\mathbf{x} \in \mathcal{D}}$. Let the sets $\mathcal{X}_\mathcal{D}$ and $\mathcal{X}_{\mathcal{D}_e}$ denote inputs in the full and erased data, respectively. Then, the ELBO can be decomposed to

$$\mathcal{L} = \sum_{\mathbf{x} \in \mathcal{X}_\mathcal{D}} \int q(\mathbf{f}_{\mathcal{X}_\mathbf{u}}|\mathcal{D}) \, p(f_\mathbf{x}|\mathbf{f}_{\mathcal{X}_\mathbf{u}}) \log p(y_\mathbf{x}|f_\mathbf{x}) \, \mathrm{d}f_\mathbf{x} \, \mathrm{d}\mathbf{f}_{\mathcal{X}_\mathbf{u}} - \mathrm{KL}[q(\mathbf{f}_{\mathcal{X}_\mathbf{u}}|\mathcal{D}) \, \| \, p(\mathbf{f}_{\mathcal{X}_\mathbf{u}})] \qquad (10)$$

where $\int p(f_\mathbf{x}|\mathbf{f}_{\mathcal{X}_\mathbf{u}}) \log p(y_\mathbf{x}|f_\mathbf{x}) \, \mathrm{d}f_\mathbf{x}$ can be evaluated in closed form [11]. To unlearn such a trained model from $\mathcal{D}_e$ ($|\mathcal{D}_e| = 100\mathrm{K}$ here), the EUBO (8) can be expressed in a similar way as the ELBO:

$$\widetilde{\mathcal{U}}_{\mathrm{adj}}(\lambda) = \sum_{\mathbf{x} \in \mathcal{X}_{\mathcal{D}_e}} \int \tilde{q}_u(\mathbf{f}_{\mathcal{X}_\mathbf{u}}|\mathcal{D}_r; \lambda) p(f_\mathbf{x}|\mathbf{f}_{\mathcal{X}_\mathbf{u}}) \log p_{\mathrm{adj}}(y_\mathbf{x}|f_\mathbf{x}; \lambda) \, \mathrm{d}f_\mathbf{x} \, \mathrm{d}\mathbf{f}_{\mathcal{X}_\mathbf{u}} + \mathrm{KL}[\tilde{q}_u(\mathbf{f}_{\mathcal{X}_\mathbf{u}}|\mathcal{D}_r; \lambda) \| q(\mathbf{f}_{\mathcal{X}_\mathbf{u}}|\mathcal{D})]$$

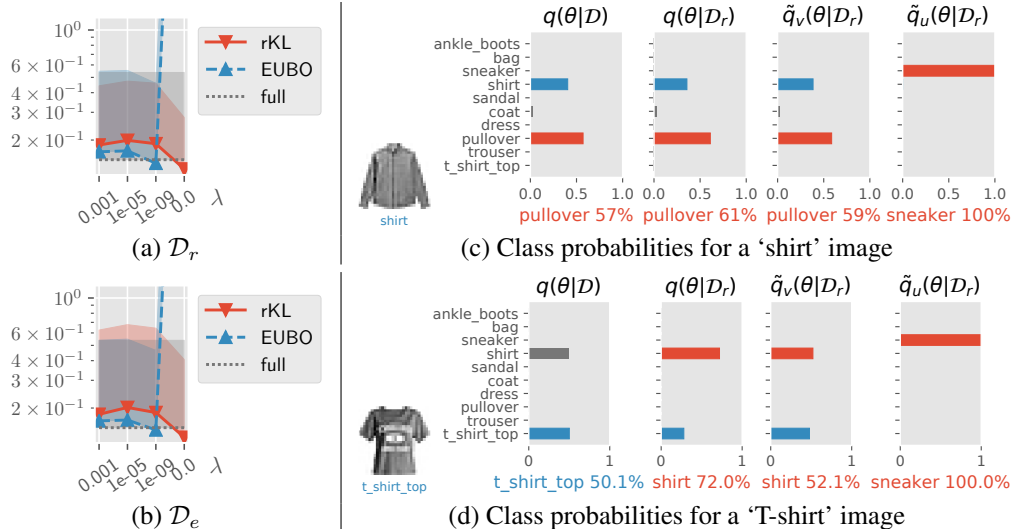

Figure 6: Graphs of averaged KL divergence vs. $\lambda$ achieved by EUBO, rKL, and $q(\boldsymbol{\theta}|\mathcal{D})$ (i.e., baseline labeled as *full*) over (a) $\mathcal{D}_r$ and (b) $\mathcal{D}_e$. (c-d) Plots of class probabilities for two images in the fashion MNIST dataset obtained using $q(\boldsymbol{\theta}|\mathcal{D})$, $q(\boldsymbol{\theta}|\mathcal{D}_r)$, optimized $\tilde{q}_v(\boldsymbol{\theta}|\mathcal{D}_r; \lambda = 0)$ and $\tilde{q}_u(\boldsymbol{\theta}|\mathcal{D}_r; \lambda = 0)$.

Table 1: KL divergence achieved by EUBO (top row) and rKL (bottom row) with varying $\lambda$ for airline dataset.

| $\lambda$ | $10^{-11}$ | $10^{-13}$ | $10^{-20}$ | $0$ |
|---|---|---|---|---|
| $\mathrm{KL}[\tilde{q}_u(\mathbf{f}_{\mathcal{X}_{\mathbf{u}}}|\mathcal{D}_r; \lambda) \parallel q(\mathbf{f}_{\mathcal{X}_{\mathbf{u}}}|\mathcal{D}_r)]$ | 2194.49 | 1943.00 | 1384.96 | 2629.71 |
| $\mathrm{KL}[\tilde{q}_v(\mathbf{f}_{\mathcal{X}_{\mathbf{u}}}|\mathcal{D}_r; \lambda) \parallel q(\mathbf{f}_{\mathcal{X}_{\mathbf{u}}}|\mathcal{D}_r)]$ | 418.42 | 367.12 | 543.45 | 455.11 |

where $p_{\mathrm{adj}}(y_{\mathbf{x}}|f_{\mathbf{x}}; \lambda) = p(y_{\mathbf{x}}|f_{\mathbf{x}})$ if $q(f_{\mathbf{x}}, \mathbf{f}_{\mathcal{X}_{\mathbf{u}}}|\mathcal{D}) > \lambda \max_{\mathbf{f}_{\mathcal{X}_{\mathbf{u}}}} q(f_{\mathbf{x}}, \mathbf{f}_{\mathcal{X}_{\mathbf{u}}}|\mathcal{D})$, and $p_{\mathrm{adj}}(y_{\mathbf{x}}|f_{\mathbf{x}}; \lambda) = 1$ otherwise. EUBO can be minimized using stochastic gradient descent with random subsets (i.e., mini-batches of size 10K) of $\mathcal{D}_e$ in each iteration. For rKL, we use the entire $\mathcal{D}_e$ in each iteration. Since $\tilde{q}_u(\mathbf{f}_{\mathcal{X}_{\mathbf{u}}}|\mathcal{D}_r; \lambda)$, $\tilde{q}_v(\mathbf{f}_{\mathcal{X}_{\mathbf{u}}}|\mathcal{D}_r; \lambda)$, and $q(\mathbf{f}_{\mathcal{X}_{\mathbf{u}}}|\mathcal{D}_r)$ in (10) [11] are all multivariate Gaussians, we can directly evaluate the performance of EUBO and rKL with varying $\lambda$ through their respective $\mathrm{KL}[\tilde{q}_u(\mathbf{f}_{\mathcal{X}_{\mathbf{u}}}|\mathcal{D}_r; \lambda) \parallel q(\mathbf{f}_{\mathcal{X}_{\mathbf{u}}}|\mathcal{D}_r)]$ and $\mathrm{KL}[\tilde{q}_v(\mathbf{f}_{\mathcal{X}_{\mathbf{u}}}|\mathcal{D}_r; \lambda) \parallel q(\mathbf{f}_{\mathcal{X}_{\mathbf{u}}}|\mathcal{D}_r)]$ which, according to Table 1, are smaller than $\mathrm{KL}[q(\mathbf{f}_{\mathcal{X}_{\mathbf{u}}}|\mathcal{D}) \parallel q(\mathbf{f}_{\mathcal{X}_{\mathbf{u}}}|\mathcal{D}_r)]$ of value 4344.09 (i.e., baseline representing no unlearning), hence demonstrating reasonable unlearning performance.

## 5 Conclusion

This paper describes novel unlearning methods for approximately unlearning a Bayesian model from a small subset of training data to be erased. Our unlearning methods are parsimonious in requiring only the approximate posterior belief of model parameters given the full data (i.e., obtained in model training with VI) and erased data to be available. This makes unlearning even more challenging due to two sources of inaccuracy in the approximate posterior belief. We introduce novel tricks of adjusted likelihood and reverse KL to curb unlearning in the region of model parameters with low approximate posterior belief where both sources of inaccuracy primarily occur. Empirical evaluations on synthetic and real-world datasets show that our proposed methods (especially reverse KL without adjusted likelihood) can effectively unlearn Bayesian models such as sparse GP and logistic regression from erased data. In practice, for the approximate posterior beliefs recovered by unlearning from erased data using our proposed methods, they can be immediately used in ML applications and continue to be improved at the same time by retraining with the remaining data at the expense of parsimony. In our future work, we will apply our our proposed methods to unlearning more sophisticated Bayesian models like the entire family of sparse GP models [5, 6, 7, 8, 16, 17, 18, 19, 20, 22, 23, 25, 31, 32, 34]) and deep GP models [33].

## Broader Impact

As discussed in our introduction (Sec. 1), a direct contribution of our work to the society in this information age is to the implementation of *personal data ownership* (i.e., enforced by the General Data Protection Regulation in the European Union [24]) by studying the problem of machine unlearning for Bayesian models. Such an implementation can boost the confidence of users about sharing their data with an application/organization when they know that the trace of their data can be reduced/erased, as requested. As a result, organizations/applications can gather more useful data from users to enhance their service back to the users and hence to the society.

Our unlearning work can also contribute to the defense against data poisoning attacks (i.e., injecting malicious training data). Instead of retraining the tampered machine learning model from scratch to recover the quality of a service, unlearning the model from the detected malicious data may incur much less time, which improves the user experience and reduces the cost due to the service disruption.

In contrast, the ability to unlearn machine learning models may also open the door to new adversarial activities. For example, in the context of data sharing, multiple parties share their data to train a common machine learning model. An unethical party can deliberately share a low-quality dataset instead of its high-quality one. After obtaining the model trained on datasets from all parties (including the low-quality dataset), the unethical party can unlearn the low-quality dataset and continue to train the model with its high-quality dataset. By doing this, the unethical party achieves a better model than other parties in the collaboration. Therefore, the possibility of machine unlearning should be considered in the design of different data sharing frameworks.

## Acknowledgments and Disclosure of Funding

This research/project is supported by the National Research Foundation, Singapore under its Strategic Capability Research Centres Funding Initiative. Any opinions, findings and conclusions or recommendations expressed in this material are those of the author(s) and do not reflect the views of National Research Foundation, Singapore.

## Footnotes

[1]A trained ML model is said to experience *catastrophic unlearning* from the erased data when its resulting performance is considerably worse than that from retraining with the remaining data.

[2]Similarly, $\mathrm{KL}[p(y|\mathcal{D}_r) \parallel q_u(y|\mathcal{D}_r)] \leq \mathrm{KL}[p(\boldsymbol{\theta}|\mathcal{D}_r) \parallel q_u(\boldsymbol{\theta}|\mathcal{D}_r)]$ holds.

[3]With a slight abuse of notation, we let $q(\boldsymbol{\theta}|\mathcal{D})$ be the approximate posterior belief that maximizes the ELBO $\mathcal{L}$ (1) (Sec. 2) from Sec. 3.3 onwards.

[4]Note that the log plots can only properly display the upper confidence interval of 1 standard deviation (shaded area) and hence do not show the lower confidence interval.

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
