[Supplementary Material]

# A Proof of Proposition 1

We first follow the proof of the log-sum inequality to prove the following inequality:

$$q_u(y|\mathcal{D}_r) \, \log \frac{q_u(y|\mathcal{D}_r)}{p(y|\mathcal{D}_r)} \leq \int q_u(\boldsymbol{\theta}|\mathcal{D}_r) \, p(y|\boldsymbol{\theta}) \, \log \frac{q_u(\boldsymbol{\theta}|\mathcal{D}_r)}{p(\boldsymbol{\theta}|\mathcal{D}_r)} \, \mathrm{d}\boldsymbol{\theta} \tag{11}$$

where $q_u(y|\mathcal{D}_r) \triangleq \mathbb{E}_{q_u(\boldsymbol{\theta}|\mathcal{D}_r)}[p(y|\boldsymbol{\theta})] = \int q_u(\boldsymbol{\theta}|\mathcal{D}_r) \, p(y|\boldsymbol{\theta}) \, \mathrm{d}\boldsymbol{\theta}$ and $p(y|\mathcal{D}_r) \triangleq \mathbb{E}_{p(\boldsymbol{\theta}|\mathcal{D}_r)}[p(y|\boldsymbol{\theta})] = \int p(\boldsymbol{\theta}|\mathcal{D}_r) \, p(y|\boldsymbol{\theta}) \, \mathrm{d}\boldsymbol{\theta}$.

*Proof.* Define the function $f(t) \triangleq t \log t$ which is convex. Then,

$$\int q_u(\boldsymbol{\theta}|\mathcal{D}_r) \, p(y|\boldsymbol{\theta}) \, \log \frac{q_u(\boldsymbol{\theta}|\mathcal{D}_r)}{p(\boldsymbol{\theta}|\mathcal{D}_r)} \, \mathrm{d}\boldsymbol{\theta}$$

$$= \int p(\boldsymbol{\theta}|\mathcal{D}_r) \, p(y|\boldsymbol{\theta}) \, f\left(\frac{q_u(\boldsymbol{\theta}|\mathcal{D}_r)}{p(\boldsymbol{\theta}|\mathcal{D}_r)}\right) \, \mathrm{d}\boldsymbol{\theta}$$

$$= \mathbb{E}_{p(\boldsymbol{\theta}|\mathcal{D}_r)}[p(y|\boldsymbol{\theta})] \int \frac{p(\boldsymbol{\theta}|\mathcal{D}_r) \, p(y|\boldsymbol{\theta})}{\mathbb{E}_{p(\boldsymbol{\theta}|\mathcal{D}_r)}[p(y|\boldsymbol{\theta})]} \, f\left(\frac{q_u(\boldsymbol{\theta}|\mathcal{D}_r)}{p(\boldsymbol{\theta}|\mathcal{D}_r)}\right) \, \mathrm{d}\boldsymbol{\theta}$$

$$\geq \mathbb{E}_{p(\boldsymbol{\theta}|\mathcal{D}_r)}[p(y|\boldsymbol{\theta})] \, f\left(\int \frac{p(\boldsymbol{\theta}|\mathcal{D}_r) \, p(y|\boldsymbol{\theta})}{\mathbb{E}_{p(\boldsymbol{\theta}|\mathcal{D}_r)}[p(y|\boldsymbol{\theta})]} \frac{q_u(\boldsymbol{\theta}|\mathcal{D}_r)}{p(\boldsymbol{\theta}|\mathcal{D}_r)} \, \mathrm{d}\boldsymbol{\theta}\right)$$

$$= \mathbb{E}_{p(\boldsymbol{\theta}|\mathcal{D}_r)}[p(y|\boldsymbol{\theta})] \, f\left(\int \frac{p(y|\boldsymbol{\theta}) \, q_u(\boldsymbol{\theta}|\mathcal{D}_r)}{\mathbb{E}_{p(\boldsymbol{\theta}|\mathcal{D}_r)}[p(y|\boldsymbol{\theta})]} \, \mathrm{d}\boldsymbol{\theta}\right)$$

$$= \mathbb{E}_{p(\boldsymbol{\theta}|\mathcal{D}_r)}[p(y|\boldsymbol{\theta})] \, f\left(\frac{\mathbb{E}_{q_u(\boldsymbol{\theta}|\mathcal{D}_r)}[p(y|\boldsymbol{\theta})]}{\mathbb{E}_{p(\boldsymbol{\theta}|\mathcal{D}_r)}[p(y|\boldsymbol{\theta})]}\right)$$

$$= \mathbb{E}_{q_u(\boldsymbol{\theta}|\mathcal{D}_r)}[p(y|\boldsymbol{\theta})] \, \log \frac{\mathbb{E}_{q_u(\boldsymbol{\theta}|\mathcal{D}_r)}[p(y|\boldsymbol{\theta})]}{\mathbb{E}_{p(\boldsymbol{\theta}|\mathcal{D}_r)}[p(y|\boldsymbol{\theta})]}$$

$$= q_u(y|\mathcal{D}_r) \, \log \frac{q_u(y|\mathcal{D}_r)}{p(y|\mathcal{D}_r)}$$

where the inequality is due to Jensen's inequality. $\qquad\square$

Then, integrating both sides of (11) w.r.t. $y$,

$$\int q_u(y|\mathcal{D}_r) \, \log \frac{q_u(y|\mathcal{D}_r)}{p(y|\mathcal{D}_r)} \, \mathrm{d}y \leq \int \int q_u(\boldsymbol{\theta}|\mathcal{D}_r) \, p(y|\boldsymbol{\theta}) \, \log \frac{q_u(\boldsymbol{\theta}|\mathcal{D}_r)}{p(\boldsymbol{\theta}|\mathcal{D}_r)} \, \mathrm{d}\boldsymbol{\theta} \, \mathrm{d}y$$

$$\int q_u(y|\mathcal{D}_r) \, \log \frac{q_u(y|\mathcal{D}_r)}{p(y|\mathcal{D}_r)} \, \mathrm{d}y \leq \int q_u(\boldsymbol{\theta}|\mathcal{D}_r) \left(\int p(y|\boldsymbol{\theta}) \, \mathrm{d}y\right) \log \frac{q_u(\boldsymbol{\theta}|\mathcal{D}_r)}{p(\boldsymbol{\theta}|\mathcal{D}_r)} \, \mathrm{d}\boldsymbol{\theta}$$

$$\int q_u(y|\mathcal{D}_r) \, \log \frac{q_u(y|\mathcal{D}_r)}{p(y|\mathcal{D}_r)} \, \mathrm{d}y \leq \int q_u(\boldsymbol{\theta}|\mathcal{D}_r) \log \frac{q_u(\boldsymbol{\theta}|\mathcal{D}_r)}{p(\boldsymbol{\theta}|\mathcal{D}_r)} \, \mathrm{d}\boldsymbol{\theta}$$

$$\mathrm{KL}[q_u(y|\mathcal{D}_r) \, \| \, p(y|\mathcal{D}_r)] \leq \mathrm{KL}[q_u(\boldsymbol{\theta}|\mathcal{D}_r) \, \| \, p(\boldsymbol{\theta}|\mathcal{D}_r)] \, .$$

# B Proof of Proposition 2

From (2),

$$\log p(\mathcal{D}_e|\mathcal{D}_r) = \log \frac{p(\mathcal{D}_e|\boldsymbol{\theta}) \, p(\boldsymbol{\theta}|\mathcal{D}_r)}{p(\boldsymbol{\theta}|\mathcal{D})}$$

$$= \log \frac{q_u(\boldsymbol{\theta}|\mathcal{D}_r) \, p(\mathcal{D}_e|\boldsymbol{\theta}) \, p(\boldsymbol{\theta}|\mathcal{D}_r)}{q_u(\boldsymbol{\theta}|\mathcal{D}_r) \, p(\boldsymbol{\theta}|\mathcal{D})}$$

$$= \log p(\mathcal{D}_e|\boldsymbol{\theta}) + \log \frac{q_u(\boldsymbol{\theta}|\mathcal{D}_r)}{p(\boldsymbol{\theta}|\mathcal{D})} - \log \frac{q_u(\boldsymbol{\theta}|\mathcal{D}_r)}{p(\boldsymbol{\theta}|\mathcal{D}_r)} \, .$$

Then, taking an expectation of both sides w.r.t. $q_u(\boldsymbol{\theta}|\mathcal{D}_r)$,

$$\log p(\mathcal{D}_e|\mathcal{D}_r) = \int q_u(\boldsymbol{\theta}|\mathcal{D}_r) \log p(\mathcal{D}_e|\boldsymbol{\theta}) \, \mathrm{d}\boldsymbol{\theta} + \int q_u(\boldsymbol{\theta}|\mathcal{D}_r) \log \frac{q_u(\boldsymbol{\theta}|\mathcal{D}_r)}{p(\boldsymbol{\theta}|\mathcal{D})} \, \mathrm{d}\boldsymbol{\theta} - \int q_u(\boldsymbol{\theta}|\mathcal{D}_r) \log \frac{q_u(\boldsymbol{\theta}|\mathcal{D}_r)}{p(\boldsymbol{\theta}|\mathcal{D}_r)} \, \mathrm{d}\boldsymbol{\theta}$$

$$= \int q_u(\boldsymbol{\theta}|\mathcal{D}_r) \, \log p(\mathcal{D}_e|\boldsymbol{\theta}) \, \mathrm{d}\boldsymbol{\theta} + \mathrm{KL}[q_u(\boldsymbol{\theta}|\mathcal{D}_r) \parallel p(\boldsymbol{\theta}|\mathcal{D})] - \mathrm{KL}[q_u(\boldsymbol{\theta}|\mathcal{D}_r) \parallel p(\boldsymbol{\theta}|\mathcal{D}_r)]$$

$$= \mathcal{U} - \mathrm{KL}[q_u(\boldsymbol{\theta}|\mathcal{D}_r) \parallel p(\boldsymbol{\theta}|\mathcal{D}_r)] \ .$$

Therefore,

$$\mathcal{U} = \log p(\mathcal{D}_e|\mathcal{D}_r) + \mathrm{KL}[q_u(\boldsymbol{\theta}|\mathcal{D}_r) \parallel p(\boldsymbol{\theta}|\mathcal{D}_r)] \geq \log p(\mathcal{D}_e|\mathcal{D}_r)$$

since $\mathrm{KL}[q_u(\boldsymbol{\theta}|\mathcal{D}_r) \parallel p(\boldsymbol{\theta}|\mathcal{D}_r)] \geq 0$. So, $\mathcal{U}$ is an upper bound of $\log p(\mathcal{D}_e|\mathcal{D}_r)$.

## C  Bayesian Linear Regression

We perform unlearning of a simple Bayesian linear regression model: $y_x = ax^3 + bx^2 + cx + d + \epsilon$ where $a = 2, b = -3, c = 1$, and $d = 0$ are the model parameters $\boldsymbol{\theta}$, and the noise is $\epsilon \sim \mathcal{N}(0, 0.05^2)$. Though the exact posterior belief of $\boldsymbol{\theta}$ is known to be a multivariate Gaussian, we choose to use a low-rank approximation (i.e., multivariate Gaussian with a diagonal covariance matrice) and represent the approximate posterior beliefs of the model parameters with independent Gaussians so that the approximation is not exact.

Fig. 7a shows the remaining data $\mathcal{D}_r$ and erased data $\mathcal{D}_e$. Note that the erased data $\mathcal{D}_e$ is informative to the approximate posterior beliefs of the model parameters $\boldsymbol{\theta}$ as $\mathcal{D}_e$ are clustered. So, the difference between the samples drawn from predictive distributions $q(y_x|\mathcal{D})$ (Fig. 7b) vs. $q(y_x|\mathcal{D}_r)$ (Fig. 7c) is large.

Table 2: KL divergences achieved by EUBO (left column) and rKL (right column) with varying $\lambda$ for synthetic linear regression dataset.

| $\lambda$ | $\mathrm{KL}[\tilde{q}_u(\boldsymbol{\theta}|\mathcal{D}_r; \lambda) \parallel q(\boldsymbol{\theta}|\mathcal{D}_r)]$ | $\mathrm{KL}[\tilde{q}_v(\boldsymbol{\theta}|\mathcal{D}_r; \lambda) \parallel q(\boldsymbol{\theta}|\mathcal{D}_r)]$ |
|---|---|---|
| 0.5 | 0.1143 | 0.1012 |
| 0.1 | 0.0899 | 0.0600 |
| 0.0 | 266.68 | 0.0158 |

From Table 2, the KL divergences achieved by EUBO and rKL with $\lambda = 0.1, 0.5$ are smaller than $\mathrm{KL}[q(\boldsymbol{\theta}|\mathcal{D}) \parallel q(\boldsymbol{\theta}|\mathcal{D}_r)]$ of value 0.1170 (i.e., baseline representing no unlearning), hence demonstrating reasonable unlearning performance. When $\lambda = 0$, EUBO suffers from catastrophic unlearning, but rKL does not. The KL divergences in Table 2 also agree with the plots of samples drawn from the predictive distributions induced by EUBO and rKL in Fig. 7 by comparing with the samples drawn from the predictive distribution obtained using VI from retraining with $\mathcal{D}_r$ in Fig. 7c.

## D  Bimodal Posterior Belief

Let the posterior belief of model parameter $\theta$ given full data $\mathcal{D}$ be a Gaussian mixture (i.e., a bimodal distribution):

$$p(\theta|\mathcal{D}) \triangleq 0.5 \, \phi(\theta; 0, 1) + 0.5 \, \phi(\theta; 2, 1) \qquad (12)$$

where $\phi(\theta; \mu, \sigma^2)$ is a Gaussian p.d.f. with mean $\mu$ and variance $\sigma^2$. We deliberately choose the likelihood of the erased data $\mathcal{D}_e$ to be

$$p(\mathcal{D}_e|\theta) \triangleq 1 + \frac{\phi(\theta; 2, 1)}{\phi(\theta; 0, 1)} \qquad (13)$$

so that the posterior belief of $\theta$ given the remaining data $\mathcal{D}_r$ is a Gaussian:

$$p(\theta|\mathcal{D}_r) \propto \frac{p(\theta|\mathcal{D})}{p(\mathcal{D}_e|\theta)} = \phi(\theta; 0, 1) \qquad (14)$$

(a) Dataset      (b) Samples from $q(y_x|\mathcal{D})$      (c) Samples from $q(y_x|\mathcal{D}_r)$

(d) EUBO with $\lambda = 0.5$      (e) EUBO with $\lambda = 0.1$      (f) EUBO with $\lambda = 0$

(g) rKL with $\lambda = 0.5$      (h) rKL with $\lambda = 0.1$      (i) rKL with $\lambda = 0$

Figure 7: Plots of (a) synthetic linear regression dataset with erased data $\mathcal{D}_e$ (crosses) and remaining data $\mathcal{D}_r$ (dots), and samples from predictive distributions obtained using VI from (b) training with full data $\mathcal{D}$ and (c) retraining with $\mathcal{D}_r$. Plots of samples from predictive distributions (d-f) $\tilde{q}_u(y_x|\mathcal{D}_r)$ and (g-i) $\tilde{q}_v(y_x|\mathcal{D}_r)$ induced, respectively, by EUBO and rKL with varying $\lambda$.

where the proportionality is due to (2).

We assume to only have access to the likelihood of the erased data in (13); the exact posterior beliefs of $\theta$ given the full data (12) and that given the remaining data (14) are not available. Instead, we have access to an approximate posterior belief $q(\theta|\mathcal{D})$ given the full data obtained using VI by minimizing $\mathrm{KL}[q(\theta|\mathcal{D}) \parallel p(\theta|\mathcal{D})]$ or, equivalently, maximizing the ELBO (Section 2):

$$q(\theta|\mathcal{D}) = \phi(\theta; 1.004, 1.390^2) . \tag{15}$$

Given the likelihood $p(\mathcal{D}_e|\theta)$ of the erased data in (13) and the approximate posterior belief $q(\theta|\mathcal{D})$ given the full data (15), unlearning from $\mathcal{D}_e$ is performed using EUBO and rKL to obtain

$$\tilde{q}_u(\theta|\mathcal{D}_r; \lambda = 0) = \phi(\theta; 0.060, 1.000^2) \quad \text{and} \quad \tilde{q}_v(\theta|\mathcal{D}_r; \lambda = 0) = \phi(\theta; 0.062, 1.018^2) ,$$

respectively. Hence, both EUBO and rKL perform reasonably well since their respective $\tilde{q}_u(\theta|\mathcal{D}_r; \lambda = 0)$ and $\tilde{q}_v(\theta|\mathcal{D}_r; \lambda = 0)$ are close to $p(\theta|\mathcal{D}_r) = \phi(\theta; 0, 1)$ (14) when $p(\theta|\mathcal{D})$ is a bimodal distribution.

# E    Gaussian Process (GP) Classification with Synthetic Moon Dataset: Additional Details and Experimental Results

This section discusses the sparse GP model that is used in the classification of the synthetic moon dataset in Sec. 4.1. Let $y_{\mathbf{x}} \in \{0, 1\}$ be the class label of $\mathbf{x} \in \mathcal{X} \subset \mathbb{R}^2$; $y_{\mathbf{x}} = 1$ denotes the 'blue' class plotted as blue dots in Fig. 4a. The probability of $y_{\mathbf{x}}$ is defined as follows:

$$\begin{aligned} p(y_{\mathbf{x}} = 1|f_{\mathbf{x}}) &\triangleq \frac{1}{1 + \exp(f_{\mathbf{x}})} \\ p(y_{\mathbf{x}} = 0|f_{\mathbf{x}}) &\triangleq \frac{\exp(f_{\mathbf{x}})}{1 + \exp(f_{\mathbf{x}})} \end{aligned} \tag{16}$$

| (a) $\mu_{\mathbf{x}|\mathcal{D}}$ | (b) $\sigma^2_{\mathbf{x}|\mathcal{D}}$ | (c) $\mu_{\mathbf{x}|\mathcal{D}_r}$ | (d) $\sigma^2_{\mathbf{x}|\mathcal{D}_r}$ |

Figure 8: Plots of approximate posterior beliefs (a-b) $q(f_{\mathbf{x}}|\mathcal{D})$ and (c-d) $q(f_{\mathbf{x}}|\mathcal{D}_r)$.

where $f_{\mathbf{x}}$ is modeled using a GP [28], that is, every finite subset of $\{f_{\mathbf{x}}\}_{\mathbf{x}\in\mathcal{X}}$ follows a multivariate Gaussian distribution. A GP is fully specified by its *prior* mean (i.e., assumed to be $0$ w.l.o.g.) and covariance $k_{\mathbf{x}\mathbf{x}'} \triangleq \mathrm{cov}(\mathbf{x}, \mathbf{x}')$, the latter of which can be defined by the widely-used squared exponential covariance function $k_{\mathbf{x}\mathbf{x}'} \triangleq \sigma^2_f \exp(-0.5\|\Lambda(\mathbf{x} - \mathbf{x}')\|^2_2)$ where $\Lambda = \mathrm{diag}[\lambda_1, \lambda_2]$ and $\sigma^2_f$ are the length-scale and signal variance hyperparameters, respectively. In this experiment, we set $\lambda_1 = 1.56$, $\lambda_2 = 1.35$, and $\sigma^2_f = 4.74$.

We employ a sparse GP model, namely, the *deterministic training conditional* (DTC) [27] approximation of the GP model with a set $\mathcal{X}_u$ of 20 *inducing inputs*. These inducing inputs are randomly selected from $\mathcal{X}$ and remain the same (and fixed) for both model training and unlearning. Given the latent function values (i.e., also known as *inducing variables*) $\mathbf{f}_{\mathcal{X}_u} \triangleq (f_{\mathbf{x}})^{\top}_{\mathbf{x}\in\mathcal{X}_u}$ at these inducing inputs, the posterior belief of the latent function value $f_{\mathbf{x}}$ at a new input $\mathbf{x}$ is a Gaussian $p(f_{\mathbf{x}}|\mathbf{f}_{\mathcal{X}_u}) = \mathcal{N}(\mathbf{k}_{\mathbf{x}\mathcal{X}_u}\mathbf{K}^{-1}_{\mathcal{X}_u\mathcal{X}_u}\mathbf{f}_{\mathcal{X}_u}, k_{\mathbf{x}\mathbf{x}} - \mathbf{k}_{\mathbf{x}\mathcal{X}_u}\mathbf{K}^{-1}_{\mathcal{X}_u\mathcal{X}_u}\mathbf{k}_{\mathcal{X}_u\mathbf{x}})$ where $\mathbf{k}_{\mathbf{x}\mathcal{X}_u} \triangleq (k_{\mathbf{x}\mathbf{x}'})_{\mathbf{x}'\in\mathcal{X}_u}$, $\mathbf{k}_{\mathcal{X}_u\mathbf{x}} = \mathbf{k}^{\top}_{\mathbf{x}\mathcal{X}_u}$, and $\mathbf{K}_{\mathcal{X}_u\mathcal{X}_u} = (k_{\mathbf{x}\mathbf{x}'})_{\mathbf{x},\mathbf{x}'\in\mathcal{X}_u}$.

Using $p(f_{\mathbf{x}}|\mathbf{f}_{\mathcal{X}_u})$ and $q(\mathbf{f}_{\mathcal{X}_u}|\mathcal{D}) \triangleq \mathcal{N}(\boldsymbol{\mu}_{\mathcal{X}_u}, \boldsymbol{\Sigma}_{\mathcal{X}_u})$, it can be derived that the approximate posterior belief $q(f_{\mathbf{x}}|\mathcal{D})$ of $f_{\mathbf{x}}$ given full data $\mathcal{D}$ is also a Gaussian with the following respective *posterior* mean and variance:

$$\mu_{\mathbf{x}|\mathcal{D}} \triangleq \mathbf{k}_{\mathbf{x}\mathcal{X}_u}\mathbf{K}^{-1}_{\mathcal{X}_u\mathcal{X}_u}\boldsymbol{\mu}_{\mathcal{X}_u} , \tag{17}$$

$$\sigma^2_{\mathbf{x}|\mathcal{D}} \triangleq k_{\mathbf{x}\mathbf{x}} - \mathbf{k}_{\mathbf{x}\mathcal{X}_u}\mathbf{K}^{-1}_{\mathcal{X}_u\mathcal{X}_u}\mathbf{k}_{\mathcal{X}_u\mathbf{x}} + \mathbf{k}_{\mathbf{x}\mathcal{X}_u}\mathbf{K}^{-1}_{\mathcal{X}_u\mathcal{X}_u}\boldsymbol{\Sigma}_{\mathcal{X}_u}\mathbf{K}^{-1}_{\mathcal{X}_u\mathcal{X}_u}\mathbf{k}_{\mathcal{X}_u\mathbf{x}} . \tag{18}$$

The approximate posterior belief $q(f_{\mathbf{x}}|\mathcal{D}_r)$ of $f_{\mathbf{x}}$ from retraining with remaining data $\mathcal{D}_r$ using VI (specifically, using $q(\mathbf{f}_{\mathcal{X}_u}|\mathcal{D}_r)$) can be derived in the same way as that of $q(f_{\mathbf{x}}|\mathcal{D})$.

The parameters $\boldsymbol{\mu}_{\mathcal{X}_u}$, $\boldsymbol{\Sigma}_{\mathcal{X}_u}$ of the approximate posterior belief $q(\mathbf{f}_{\mathcal{X}_u}|\mathcal{D})$ is optimized by maximizing the ELBO with stochastic gradient ascent (let $\boldsymbol{\theta} = \mathbf{f}_{\mathcal{X}_u}$ in (1) in Sec. 2):

$$\mathbb{E}_{\mathbf{f}_{\mathcal{X}_u}\sim q(\mathbf{f}_{\mathcal{X}_u}|\mathcal{D})}\left[\log p(\mathcal{D}|\mathbf{f}_{\mathcal{X}_u}) - \log q(\mathbf{f}_{\mathcal{X}_u}|\mathcal{D}) + \log p(\mathbf{f}_{\mathcal{X}_u})\right]$$

where $p(\mathcal{D}|\mathbf{f}_{\mathcal{X}_u})$ is computed using (16), (17) and (18).

Fig. 8 visualizes $q(f_{\mathbf{x}}|\mathcal{D})$ (Figs. 8a and 8b) and $q(f_{\mathbf{x}}|\mathcal{D}_r)$ (Figs. 8c and 8d) whose corresponding predictive distributions $q(y_{\mathbf{x}} = 1|\mathcal{D})$ and $q(y_{\mathbf{x}} = 1|\mathcal{D}_r)$ are shown in Figs. 4b and 4c, respectively. On the other hand, Figs. 9 and 10 visualize the approximate posterior beliefs $\tilde{q}_u(f_{\mathbf{x}}|\mathcal{D}_r; \lambda)$ and $\tilde{q}_v(f_{\mathbf{x}}|\mathcal{D}_r; \lambda)$ induced, respectively, by EUBO and rKL whose corresponding predictive distributions $\tilde{q}_u(y_{\mathbf{x}} = 1|\mathcal{D}_r)$ and $\tilde{q}_v(y_{\mathbf{x}} = 1|\mathcal{D}_r)$ are shown in Figs. 4f-k. Similar to the comparison between predictive distributions $\tilde{q}_u(y_{\mathbf{x}} = 1|\mathcal{D}_r)$ vs. $q(y_{\mathbf{x}} = 1|\mathcal{D}_r)$ in Sec. 4.1, it can be observed that the approximate posterior belief $\tilde{q}_u(f_{\mathbf{x}}|\mathcal{D}_r; \lambda = 10^{-9})$ induced by EUBO is similar to $q(f_{\mathbf{x}}|\mathcal{D}_r)$ obtained using VI from retraining with $\mathcal{D}_r$ (compare Figs. 9c vs. 8c and Figs. 9d vs. 8d). However, $\tilde{q}_u(f_{\mathbf{x}}|\mathcal{D}_r; \lambda = 0)$ induced by EUBO differs from $q(f_{\mathbf{x}}|\mathcal{D}_r)$ obtained using VI from retraining with $\mathcal{D}_r$ (compare Figs. 9e vs. 8c and Figs. 9f vs. 8d). On the other hand, both the approximate posterior beliefs $\tilde{q}_v(f_{\mathbf{x}}|\mathcal{D}_r; \lambda = 10^{-9})$ and $\tilde{q}_v(f_{\mathbf{x}}|\mathcal{D}_r; \lambda = 0)$ induced by rKL are similar to $q(f_{\mathbf{x}}|\mathcal{D}_r)$ obtained using VI from retraining with $\mathcal{D}_r$ (compare Fig. 10 vs. Figs. 8c-d).

## F  A Note on Erasing Informative Data

In this section, we study the performance of our unlearning methods when erasing a large quantity of data or with different distributions of erased data (i.e., erasing the data randomly vs. deliberately

(a) Mean of $\tilde{q}_u(f_{\mathbf{x}}|\mathcal{D}_r; \lambda = 10^{-5})$     (b) Variance of $\tilde{q}_u(f_{\mathbf{x}}|\mathcal{D}_r; \lambda = 10^{-5})$

(c) Mean of $\tilde{q}_u(f_{\mathbf{x}}|\mathcal{D}_r; \lambda = 10^{-9})$     (d) Variance of $\tilde{q}_u(f_{\mathbf{x}}|\mathcal{D}_r; \lambda = 10^{-9})$

(e) Mean of $\tilde{q}_u(f_{\mathbf{x}}|\mathcal{D}_r; \lambda = 0)$     (f) Variance of $\tilde{q}_u(f_{\mathbf{x}}|\mathcal{D}_r; \lambda = 0)$

Figure 9: Plots of approximate posterior belief $\tilde{q}_u(f_{\mathbf{x}}|\mathcal{D}_r; \lambda)$ induced by EUBO for varying $\lambda$.

(a) Mean of $\tilde{q}_v(f_{\mathbf{x}}|\mathcal{D}_r; \lambda = 10^{-5})$     (b) Variance of $\tilde{q}_v(f_{\mathbf{x}}|\mathcal{D}_r; \lambda = 10^{-5})$

(c) Mean of $\tilde{q}_v(f_{\mathbf{x}}|\mathcal{D}_r; \lambda = 10^{-9})$     (d) Variance of $\tilde{q}_v(f_{\mathbf{x}}|\mathcal{D}_r; \lambda = 10^{-9})$

(e) Mean of $\tilde{q}_v(f_{\mathbf{x}}|\mathcal{D}_r; \lambda = 0)$     (f) Variance of $\tilde{q}_v(f_{\mathbf{x}}|\mathcal{D}_r; \lambda = 0)$

Figure 10: Plots of approximate posterior belief $\tilde{q}_v(f_{\mathbf{x}}|\mathcal{D}_r; \lambda)$ induced by rKL for varying $\lambda$.

erasing all data in a given class). Let us consider the experiment in Sec. 4.1 on the sparse GP model (i.e., the model parameters $\boldsymbol{\theta}$ in (1) in Sec. 2 are inducing variables $\mathbf{f}_{\mathcal{X}_u}$) in the classification of the synthetic moon dataset as it allows us to easily visualize both the approximate posterior beliefs of the latent function $f_{\mathbf{x}}$ and the predictive distributions of the output/observation $y_{\mathbf{x}}$. A key factor influencing the performance of our unlearning methods in the above-mentioned scenarios is the difference between the approximate posterior belief of model parameters $\mathbf{f}_{\mathcal{X}_u}$ given remaining data $\mathcal{D}_r$ vs. that given full data $\mathcal{D}$. We quantify such a difference by how much the erased data $\mathcal{D}_e$ reduces the entropy of model parameters/inducing variables $\mathbf{f}_{\mathcal{X}_u}$ given remaining data $\mathcal{D}_r$:

$$\mathcal{I} \triangleq H(\mathbf{f}_{\mathcal{X}_u}|\mathcal{D}_r) - H(\mathbf{f}_{\mathcal{X}_u}|\mathcal{D}) = -\int q(\mathbf{f}_{\mathcal{X}_u}|\mathcal{D}_r)\log q(\mathbf{f}_{\mathcal{X}_u}|\mathcal{D}_r)\,\mathrm{d}\mathbf{f}_{\mathcal{X}_u} + \int q(\mathbf{f}_{\mathcal{X}_u}|\mathcal{D})\log q(\mathbf{f}_{\mathcal{X}_u}|\mathcal{D})\,\mathrm{d}\mathbf{f}_{\mathcal{X}_u}\,.$$
(19)

Note that $\mathcal{I}$ (19) is not the same as the mutual information (i.e., information gain) between $\mathbf{f}_{\mathcal{X}_u}$ and $\mathbf{y}_{\mathcal{D}_e} \triangleq (y_{\mathbf{x}})_{(\mathbf{x},y_{\mathbf{x}})\in\mathcal{D}_e}^{\top}$ given $\mathcal{D}_r$, which is equal to $H(\mathbf{f}_{\mathcal{X}_u}|\mathcal{D}_r) - \mathbb{E}_{p(\mathbf{y}_{\mathcal{D}_e}|\mathcal{D}_r)}[H(\mathbf{f}_{\mathcal{X}_u}|\mathcal{D}_r, \mathbf{y}_{\mathcal{D}_e})]$ with an expensive-to-evaluate expectation term. Furthermore, the outputs/observations $\mathbf{y}_{\mathcal{D}_e}$ are known from $\mathcal{D}_e$. These therefore prompt us to choose $\mathcal{I}$ (19) as the measure of how much the erased data $\mathcal{D}_e$ reduces the entropy of model parameters/inducing variables $\mathbf{f}_{\mathcal{X}_u}$ given remaining data $\mathcal{D}_r$.

We investigate 4 different scenarios in the order of increasing $\mathcal{I}$:

1. Randomly selected $\mathcal{D}_e$ ($\mathcal{I} = 0.27$): The erased data of size $|\mathcal{D}_e| = 20$ are randomly selected from $\mathcal{D}$. Hence, they are not necessarily near the decision boundary, i.e., $\mathcal{D}_e$ does not reduce the entropy of model parameters/inducing variables $\mathbf{f}_{\mathcal{X}_u}$ given $\mathcal{D}_r$ much;

2. Partially 'yellow' $\mathcal{D}_e$ ($\mathcal{I} = 1.59$): The erased data of size $|\mathcal{D}_e| = 30$ are labeled with the 'yellow' class and comprise inputs $\mathbf{x}$ with the largest possible first component $x_0$. Such a choice ensures that the erased data group together to cover a part of the decision boundary, as shown in Fig. 11d;

3. Largely 'yellow' $\mathcal{D}_e$ ($\mathcal{I} = 2.06$): The erased data of size $|\mathcal{D}_e| = 40$ are labeled with the yellow class and comprise inputs $\mathbf{x}$ with the largest possible first component $x_0$. As the quantity of the erased data $\mathcal{D}_e$ increases from 30 (i.e., partially 'yellow' $\mathcal{D}_e$) to 40, $\mathcal{D}_e$ covers a larger part of the decision boundary (compare Figs. 11g vs. 11d); and

4. Fully 'yellow' $\mathcal{D}_e$ ($\mathcal{I} = 3.86$): The erased data of size $|\mathcal{D}_e| = 50$ comprise all data in the yellow class. In this case, $\mathcal{D}_e$ reduces the entropy of the model parameters/inducing variables $\mathbf{f}_{\mathcal{X}_u}$ given $\mathcal{D}_r$ the most when compared to the above 3 scenarios.

As $\mathcal{I}$ increases, the difference between the approximate posterior belief of $\mathbf{f}_{\mathcal{X}_u}$ given remaining data $\mathcal{D}_r$ vs. that given full data $\mathcal{D}$ increases. Though it is difficult to visualize such a difference directly, Proposition 1 tells us that this difference can be alternatively understood by comparing the predictive distributions $q(y_{\mathbf{x}} = 1|\mathcal{D}_r)$ in Table 3 vs. $q(y_{\mathbf{x}} = 1|\mathcal{D})$ in Fig. 4b.

Fig. 11 shows results of averaged KL divergences (i.e., performance metric described in Sec. 4) achieved by EUBO, rKL, and $q(\mathbf{f}_{\mathcal{X}_u}|\mathcal{D})$ over $\mathcal{D}_r$ and $\mathcal{D}_e$ for the 4 scenarios above. Table 3 also analyzes the performance of our unlearning methods qualitatively by plotting the means of the approximate posterior beliefs $\tilde{q}_u(f_{\mathbf{x}}|\mathcal{D}_r; \lambda)$ and $\tilde{q}_v(f_{\mathbf{x}}|\mathcal{D}_r; \lambda)$ induced, respectively, by EUBO and rKL with the corresponding predictive distributions $\tilde{q}_u(y_{\mathbf{x}} = 1|\mathcal{D}_r)$ and $\tilde{q}_v(y_{\mathbf{x}} = 1|\mathcal{D}_r)$, together with the mean of the approximate posterior belief $q(f_{\mathbf{x}}|\mathcal{D}_r)$ with the corresponding predictive distribution $q(y_{\mathbf{x}} = 1|\mathcal{D}_r)$ obtained using VI from retraining with remaining data $\mathcal{D}_r$. The following observations result:

- Fig. 11 shows that as $\mathcal{I}$ increases across the 4 scenarios, the averaged KL divergence between $q(y_{\mathbf{x}}|\mathcal{D})$ vs. $q(y_{\mathbf{x}}|\mathcal{D}_r)$ over $\mathcal{D}_r$ and $\mathcal{D}_e$ (i.e., baseline labeled as *full*) generally increases.

- In the scenario of randomly selected $\mathcal{D}_e$ (i.e., $\mathcal{I}$ is small), we expect the difference between the predictive distributions $q(y_{\mathbf{x}}|\mathcal{D})$ vs. $q(y_{\mathbf{x}}|\mathcal{D}_r)$ over $\mathcal{D}_r$ and $\mathcal{D}_e$ to be small, which is reflected in the very small averaged KL divergences of about 0.002 and 0.004 achieved by $q(\mathbf{f}_{\mathcal{X}_u}|\mathcal{D})$ (i.e., baseline labeled as *full*) in Figs. 11b and 11c, respectively. It can also be observed that though EUBO and rKL with $\lambda \in \{10^{-5}, 10^{-9}\}$ achieve smaller averaged KL divergences than that of $q(\mathbf{f}_{\mathcal{X}_u}|\mathcal{D})$ (i.e., baseline), EUBO's averaged KL divergence increases beyond than that of the baseline when $\lambda = 0$, but remains very small. As a result, the first row in Table 3 shows that when $\lambda = 10^{-9}$ or $\lambda = 0$, the predictive distributions

$\tilde{q}_u(y_{\mathbf{x}} = 1|\mathcal{D}_r)$ and $\tilde{q}_v(y_{\mathbf{x}} = 1|\mathcal{D}_r)$ induced, respectively, by EUBO and rKL are similar to $q(y_{\mathbf{x}} = 1|\mathcal{D}_r)$ obtained using VI from retraining with $\mathcal{D}_r$. Hence, we can conclude that both EUBO and rKL perform reasonably well in this scenario, even when $\lambda = 0$.

- In the scenarios of partially and largely 'yellow' $\mathcal{D}_e$, $\mathcal{I}$ is much larger than that in the scenario of randomly selected $\mathcal{D}_e$. So, we expect an increase in the difference between the predictive distributions $q(y_{\mathbf{x}}|\mathcal{D})$ vs. $q(y_{\mathbf{x}}|\mathcal{D}_r)$ over $\mathcal{D}_r$ and $\mathcal{D}_e$. It can be observed from Figs. 11e-f and 11h-i that when $\lambda = 0$, EUBO performs poorly as its averaged KL divergence is larger than that of $q(\mathbf{f}_{\mathcal{X}_u}|\mathcal{D})$ (i.e., baseline labeled as *full*), while rKL performs well as its averaged KL divergence is much smaller than that of the baseline. On the other hand, when $\lambda = 10^{-9}$, both EUBO and rKL perform well, which can also be observed from the second and third rows of Table 3. These plots also show that while the predictive distributions $\tilde{q}_v(y_{\mathbf{x}} = 1|\mathcal{D}_r)$ induced by rKL with $\lambda = 10^{-9}$ are not as similar to $q(y_{\mathbf{x}} = 1|\mathcal{D}_r)$ as $\tilde{q}_u(y_{\mathbf{x}} = 1|\mathcal{D}_r)$ induced by EUBO with $\lambda = 10^{-9}$, the performance of rKL with $\lambda = 0$ is more robust.

- In the scenario of fully 'yellow' $\mathcal{D}_e$ (i.e., $\mathcal{I}$ is largest), the difference between the predictive distributions $q(y_{\mathbf{x}}|\mathcal{D})$ vs. $q(y_{\mathbf{x}}|\mathcal{D}_r)$ over $\mathcal{D}_r$ and $\mathcal{D}_e$ is larger than that in the above 3 scenarios. Except for EUBO with $\lambda = 0$, the predictive distributions $\tilde{q}_u(y_{\mathbf{x}}|\mathcal{D}_r)$ and $\tilde{q}_v(y_{\mathbf{x}}|\mathcal{D}_r)$ induced, respectively, by EUBO and rKL are closer to $q(y_{\mathbf{x}}|\mathcal{D}_r)$ than $q(y_{\mathbf{x}}|\mathcal{D})$ as they achieve smaller averaged KL divergences than that of $q(\mathbf{f}_{\mathcal{X}_u}|\mathcal{D})$, as shown in Figs. 11k-l. However, the fourth row of Table 3 shows that both EUBO and rKL do not perform that well. Nevertheless, it can be observed that when $\lambda = 0$, the predictive distribution $\tilde{q}_v(y_{\mathbf{x}} = 1|\mathcal{D}_r)$ induced by rKL is still usable while $\tilde{q}_u(y_{\mathbf{x}} = 1|\mathcal{D}_r)$ induced by EUBO is useless.

To summarize, when only an approximate posterior belief $q(\boldsymbol{\theta}|\mathcal{D})$ of model parameters $\boldsymbol{\theta} = \mathbf{f}_{\mathcal{X}_u}$ given full data $\mathcal{D}$ (i.e., obtained in model training with VI) is available, both EUBO and rKL can perform well if the difference between the approximate posterior belief of model parameters given remaining data $\mathcal{D}_r$ vs. that given full data $\mathcal{D}$ is sufficiently small. In practice, this is expected due to the small quantity of erased data and redundancy in real-world datasets. In the case where the erased data is highly informative, the approximate posterior belief $\tilde{q}_v(\boldsymbol{\theta}|\mathcal{D}_r; \lambda = 0)$ induced by rKL remains usable by being close to $q(\boldsymbol{\theta}|\mathcal{D})$ and hence sacrificing its unlearning performance. On the other hand, EUBO may suffer from poor unlearning performance when $\lambda$ is too small.

The above remark highlights the limitation of our unlearning methods when the erased data $\mathcal{D}_e$ is informative and only the approximate posterior belief $q(\boldsymbol{\theta}|\mathcal{D})$ is available. Such a limitation is due to the lack of information about the difference between the exact posterior belief $p(\boldsymbol{\theta}|\mathcal{D})$ vs. the approximate one $q(\boldsymbol{\theta}|\mathcal{D})$ (Sec. 3.3), which motivates future investigation into maintaining additional information about this difference during the model training with VI to improve the unlearning performance. In practice, an ML application may require an unlearning method to be time-efficient in order to satisfy the constraint on the response time to a user's request for her data to be erased while not rendering the model useless (e.g., due to catastrophic unlearning). After processing the user's request, the ML application can continue to improve the approximate posterior belief recovered by unlearning from erased data (i.e., using our proposed EUBO or rKL) by retraining with the remaining data at the expense of parsimony (i.e., in terms of time and space costs).

One may wonder how our unlearning methods can handle multiple users' request arriving sequentially over time. To avoid approximation errors from accumulating, we can adopt the approach of *lazy* unlearning by aggregating all the (past and new) users' erased data into $\mathcal{D}_e$ and performing unlearning (i.e., using only $q(\boldsymbol{\theta}|\mathcal{D})$ and $\mathcal{D}_e$) as and when necessary. As expected, our unlearning methods can perform well, provided that the aggregated erased data $\mathcal{D}_e$ remains sufficiently small or contains enough redundancy.

## G  Logistic Regression with Fashion MNIST Dataset: Additional Experimental Results

In this section, we will present the following:

- Additional visualizations of the class probabilities for images in $\mathcal{D}_r$ evaluated at the mean of the approximate posterior beliefs obtained using EUBO and rKL with $\lambda = 0$ in Fig. 13, and

Figure 11: Plots of (a,d,g,j) synthetic moon dataset with erased data $\mathcal{D}_e$ (crosses) and remaining data $\mathcal{D}_r$ (dots) in 4 different scenarios. Graphs of averaged KL divergence vs. $\lambda$ achieved by EUBO, *reverse KL* (rKL), and $q(\boldsymbol{\theta}|\mathcal{D})$ (i.e., baseline labeled as *full*) over $\mathcal{D}_r$ and $\mathcal{D}_e$ in the following 4 scenarios: (b-c) randomly selected $\mathcal{D}_e$, (e-f) partially 'yellow' $\mathcal{D}_e$, (h-i) largely 'yellow' $\mathcal{D}_e$, and (k-l) fully 'yellow' $\mathcal{D}_e$.

Table 3: Plots of the mean of approximate posterior belief $q(f_{\mathbf{x}}|\mathcal{D}_r)$ with the corresponding predictive distribution $q(y_{\mathbf{x}}=1|\mathcal{D}_r)$ obtained using VI from retraining with remaining data $\mathcal{D}_r$, and also the means of approximate posterior beliefs $\tilde{q}_u(f_{\mathbf{x}}|\mathcal{D}_r;\lambda)$ and $\tilde{q}_v(f_{\mathbf{x}}|\mathcal{D}_r;\lambda)$ induced, respectively, by EUBO and rKL with the corresponding predictive distributions $\tilde{q}_u(y_{\mathbf{x}}=1|\mathcal{D}_r)$ and $\tilde{q}_v(y_{\mathbf{x}}=1|\mathcal{D}_r)$ for $\lambda \in [10^{-9},0]$. The 1-st, 2-nd, 3-rd, and 4-th rows correspond to the following 4 respective scenarios: randomly selected $\mathcal{D}_e$, partially 'yellow' $\mathcal{D}_e$, largely 'yellow' $\mathcal{D}_e$, and fully 'yellow' $\mathcal{D}_e$.

| Dataset | Retrained | | EUBO | | rKL | |
|---|---|---|---|---|---|---|
| | Mean $\mu_{\mathbf{x}|\mathcal{D}_r}$ | $q(y_{\mathbf{x}}=1|\mathcal{D}_r)$ | Mean | $\tilde{q}_u(y_{\mathbf{x}}=1|\mathcal{D}_r)$ | Mean | $\tilde{q}_v(y_{\mathbf{x}}=1|\mathcal{D}_r)$ |

$\lambda = 10^{-9}$

$\lambda = 0$

$\lambda = 10^{-9}$

$\lambda = 0$

$\lambda = 10^{-9}$

$\lambda = 0$

$\lambda = 10^{-9}$

$\lambda = 0$

Figure 12: Graphs of averaged KL divergence vs. $\lambda$ achieved by EUBO, rKL, and $q(\boldsymbol{\theta}|\mathcal{D})$ (i.e., baseline labeled as *full*) over $\mathcal{D}_r$ and $\mathcal{D}_e$ for the fashion MNIST dataset. The approximate posterior beliefs of the model parameters/weights are represented by (a-b) independent Gaussians (i.e., diagonal covariance matrices) and (c-d) multivariate Gaussians (i.e., full covariance matrices).

- Comparison of the unlearning performance obtained using approximate posterior beliefs modeled with independent Gaussians (i.e., diagonal covariance matrices) vs. that modeled with multivariate Gaussians (i.e., full covariance matrices).

Fig. 13 shows the class probabilities for the images in $\mathcal{D}_r$ evaluated at the mean of the approximate posterior beliefs with $\lambda = 0$. Figs. 13a-d and 13g show that rKL induces the highest class probability for the same class as that of $q(\boldsymbol{\theta}|\mathcal{D}_r)$. In Figs. 13e-f and 13h, the class probabilities obtained using optimized $\tilde{q}_v(\boldsymbol{\theta}|\mathcal{D}_r; \lambda = 0)$ resemble that obtained using $q(\boldsymbol{\theta}|\mathcal{D})$, though the probability of the correct class is reduced due to unlearning.

Fig. 12 shows the averaged KL divergences of EUBO, rKL, and $q(\boldsymbol{\theta}|\mathcal{D})$ where the approximate posterior beliefs are modeled with independent Gaussians (i.e., diagonal covariance matrices) in Figs. 12a-b and multivariate Gaussians (i.e., full covariance matrices) in Figs. 12c-d. It can be observed that the averaged KL divergences between $q(y_{\mathbf{x}}|\mathcal{D})$ vs. $q(y_{\mathbf{x}}|\mathcal{D}_r)$ over $\mathcal{D}_r$ and $\mathcal{D}_e$ (i.e., baselines labeled as *full*) decrease when multivariate Gaussians with full covariance matrices are used to model the approximate posterior beliefs instead (compare the baselines labeled as *full* in Figs. 12c-d vs. that in Figs. 12a-b). Furthermore, in such a case, the unlearning performance of both EUBO and rKL improve as their averaged KL divergences are not as large (relative to the baselines) as that using independent Gaussians.

Figure 13: Plots of class probabilities for images in $\mathcal{D}_r$ obtained using $q(\boldsymbol{\theta}|\mathcal{D})$, $q(\boldsymbol{\theta}|\mathcal{D}_r)$, optimized $\tilde{q}_v(\boldsymbol{\theta}|\mathcal{D}_r; \lambda = 0)$ and $\tilde{q}_u(\boldsymbol{\theta}|\mathcal{D}_r; \lambda = 0)$.