[Reviews · NeurIPS 2020]

Review 1

Summary and Contributions: This paper considers the problem of machine _un_learning, where the goal is to have a trained machine learning model "unlearn" from a certain subset of the data that is identified as being malicious or that is legally retracted by a user in the context of personal data ownership regulations. Since the naive solution of retraining the model from scratch with the remaining data is practically inefficient, machine unlearning aims to exactly or approximately obtain such a model without the need to retrain, while avoiding substantial performance deterioration ("catastrophic unlearning"). The paper proposes a new Bayesian approach to unlearning whose loss function directly quantifies the approximation gap by captures the KL divergence between the approximate model posterior resulting from unlearning from the erased data and the exact model posterior from retraining with the remaining data. The authors show that minimizing this KL divergence is equivalent to minimizing an evidence _upper_ bound in a variational inference framework. To address the challenge that varatiational inference only yields an approximate model posterior, the paper proposes two tricks, namely adjusting the likelihood of the erased data, and using the reverse KL divergence. An empirical evaluation on synthetic and real-world datasets demonstrates the efficacy of the proposed approach when used with sparse Gaussian process and logistic regression models.

Strengths: The paper tackles the important open problem of machine unlearning and provides a promising and principled method rooted in Bayesian inference. The described methodology seems novel, as Bayesian inference techniques (e.g. variational inference) have not been previouly applied to tackle the machine unlearning problem, to the best of my knowledge. The provided formulation in terms of information theoretic measures of divergence between posterior distributions over model parameters is intuitively sensible, and the introduced approximations and bounds required for practical usage of this approach appear well-motivated and sound. The paper is relevant for the NeurIPS community, and provides conceptual contributions that will likely be of interest to researchers and practitioners in machine learning.

Weaknesses: The main weakness I see in this paper is with the empirical evaluation. Firstly, the proposed approach to machine unlearning is only assessed on rather small-scale problems, involving somewhat simple/small models and/or datasets. It would make the paper significantly stronger if it would consider larger scale models such as deep neural networks and accordingly more complex datasets. Given that such models are increasingly employed in important practical applications, machine unlearning seems particularly relevant for them. While there are indeed challenges in applying variational inference in the context of such larger models, I believe that recent advances in Bayesian neural networks (e.g. Blundell et al. 2015, Kingma et al. 2015, Khan et al. 2018) do provide possible ways to scale the proposed approach to such models, unless I am missing a fundamental reason why the proposed approach does not scale in this regard. Secondly, while the paper mentions previous work tackling the machine unlearning problem, there is no empirical comparison against these previous approaches, making it hard to place this work in the context of the existing literature landscape. As a result of these points, it is challenging to assess the potential practical impact and significance of this work for relevant real-world applications, and it would make the paper significantly stronger if such additional results and baselines would be considered in the experiments. References: Blundell et al., "Weight uncertainty in neural networks", ICML 2015 Kingma et al., "Variational dropout and the local reparameterization trick", NeurIPS 2015 Khan et al., "Fast and Scalable Bayesian Deep Learning by Weight-Perturbation in Adam", ICML 2018

Correctness: Yes, the claims, derivations and empirical methodology presented in the paper appear to be correct, as far as I can tell.

Clarity: The paper is fairly well written overall. There are certain parts of Section 3 which are quite dense and require a bit more effort to be parsed due to the various different notations used for the relevant concepts, but it is understandable all in all. The conceptual Figures 1 and 2 are well readable and helpful for the intuitive understanding.

Relation to Prior Work: The paper does mention the small corpus of prior work that is available in the area of machine unlearning, and briefly mentions them in relation to the proposed method. However, the paper would benefit from a more detailed discussion on the differences and similarities to such previous works.

Reproducibility: Yes

Additional Feedback: POST-REBUTTAL: Thank you for addressing some of my concerns raised. I am still in favour of seeing larger-scale experiment to showcase that this approach works in settings where it might be most important for practical applications. On the other hand, I appreciate the proposed approach and the proof-of-concept that Bayesian unlearning can work, which will serve as inspiration for upcoming works on this topic. I am thus still in favour of seeing the paper accepted, keeping my score at 6. In any case, I strongly encourage the authors to continue working on what seems to be a very promising research direction, and to take into account all feedback in order to improve their work. ============== Questions: - as you point out, minimizing the reverse KL divergence tends to overestimate the variance, which is more desirable than underestimating it; what are the disadvantages of using this formulation? are there scenarios where this might lead to issues? a brief discussion of this would be useful - what are the limitations of your approach in terms of scalability (e.g. with model and/or dataset size)? a discussion of this would be helpful as well - it seems natural to consider variational inference for estimating the posterior over model parameters in the context of your framework; how straightforward would it be to extend this to other ways of inference, such as MCMC or the Laplace approximation?


Review 2

Summary and Contributions: The paper tackles the problem of unlearning, i.e. getting the model to forget that it has seen part of the data. To do this the authors propose a VI based approach, deriving two possible approaches optimizing either KL(q||p) or KL(p||q).

Strengths: The theory is well motivated and argued for together with an extensive appendix for the claimed statements. The paper is well structured and also discussing possible drawbacks of individual parts. Concerning the relevance to the NeurIPS community: While the task of unlearning is not novel to the paper, it is still a largely unsolved problem that will only gain in importance as our interaction with models that learn from our data, but for data protection reasons also should be able to unlearn once our relationship with the company/institution employing the model ends, grows.

Weaknesses: The major motivation for unlearning is the desire to avoid having to retrain an expensive model from scratch every time part of the training data should be forgotten. However, the model is evaluated primarily on very small models (a GP on toy data, logistic regression with 5 parameters, and a small net). Such models serve very well as a proof-of-concept to show that the approach, in general, can work, but they are rather limited in demonstrating whether the model can be properly used in practice, as they are cheap enough so that retraining is an option. Whether the approach still works for more expensive models remains to be shown and the fact that even in the smallish real-world settings the simple mean-field Gaussian variational posterior struggles to unlearn seems worrisome and indicates that the approach won't be the final solution to the problem but requires further work. I still suggest acceptance as the flow-based results (Figure 4c,d) suggest that there is hope once more flexible posteriors are learned.

Correctness: Both the theoretical claims, proofs, as well as the empirical methodology are correct as far as I can tell.

Clarity: The paper is well written, with a clear storyline.

Relation to Prior Work: I am not very well versed in the unlearning literature, and cannot properly judge whether the presented prior work gives a complete overview on the current status (I have to rely on the judgment of the other reviewers for that), but it seems to be as far as I can tell. The experiments themselves are still in a status of evaluating whether the method works at all and are not yet a proper evaluation of how the method compares to other unlearning approaches and which is best for what kind of tasks.

Reproducibility: No

Additional Feedback: ##### Post Rebuttal Update: I thank the authors for their feedback. The airline experiment is a first step in showing that the method has the potential to scale (I encourage the authors to continue working in that direction). While the dependence on the quality of the posterior belief is a downside, it is a problem shared by most Bayesian approaches. I keep my score and still recommend acceptance. ########## ## Major Comments/Questions - All of the models are rather small, and even the largest, the neural net with three layers, if further constrained to modeling only the posterior of the last layer. Can the authors comment on the scalability problem that limits the practical applicability of the model? - The results on the real data show that the approach struggles a lot when the common approach of Gaussian variational posteriors is used, requiring flow-based approaches for effective unlearning. I expect this to further hurt the scalability of the approach. Can this problem be avoided? - l282 claims that the neural net results can similarly be improved by a flow-based approach as for the logistic regression. However, this claim seems to not be validated neither in the main paper nor the appendix. Did I just overlook these results and if not, can the authors show that the observed behavior in Section 4.2 generalizes to the neural net case? Or is this just an assumption so far? In the case of the latter, this should clearly be stated in the paper. - Concerning the same neural net experiment, Louizos and Welling (ICML 2017) have already been able to train a LeNet5 sized network with a flow-based variational posterior for the weights of the whole network. As such it should be directly applicable to this experiment and could be something to evaluate a more flexible posterior. ## Minor Comments/Questions - A large motivation in the introduction and broader impact sections of the paper is personal data ownership and the GDPR. Given that the method is only approximate and forgetting a specific data point cannot be theoretically guaranteed, can the authors comment on how practically applicable the proposed approach is? Or are the GDPR requirements so strict as to require retraining/proofed forgetting to be fulfilled making the paper a nice first step, but leaving lots of further problems until the formal requirements are met? - l67 and others refer to the Kullback Leibler divergence as a distance. Given that it is not a distance due to its lack of symmetry it should properly be called divergence or relative entropy. - The paper is currently lacking almost all details required to reproduce the claimed results (hyperparameters, optimizers,...). The provided code is very extensive and detailed and if the authors publish it also upon acceptance of the paper this alleviates the reproducibility problem. If not, the paper requires a detailed discussion of all such details in the appendix. Even if the code is provided such a specification in the appendix would be very helpful for the reader to get an idea of the setup without requiring him/her to work through the code. _____ Louizos and Welling, Multiplicative Normalizing Flows for Variational Bayesian Neural Networks, ICML 2017


Review 3

Summary and Contributions: The authors proposed novel approximate unlearning methods for modern Bayesian models based on the variational inference approach. Firstly, they showed that minimizing the KL-divergence between the posterior and the approximate posterior on remaining data is equal to minimize that between the predictive and approximate predictive distribution. From this result, they also claimed that minimizing KL(q_{u}[\theta|D_{r}) || p(\theta|D_{r})] is equal to minimize the evidence upper bound (EUBO), which is the counterpart of ELBO. Furthermore, to deal with inaccuracy problems due to \theta with small q(\theta|D), they proposed two tricks; one of these is Adjusted Likelihood, and the other is using reverse KL. They confirmed the reverse KL method could effectively unlearn Bayesian models through several experiments on synthetic and real-world datasets.

Strengths: ・The idea is somewhat novel; to the best of my knowledge, the first paper uses a variational approach to the Bayesian unlearning scheme. ・The authors confirmed the performances and usefulness of the proposed method through several experiments and various hyperparameter settings (but I mentioned below, the more results, e.g., the optimization path, are necessary to confirm the characteristics of the proposed method).

Weaknesses: Although the idea itself is somewhat novel, there are some unclear points in this paper. (a) : The lack of theoretical guarantees on EUBO There are no theoretical guarantees for the properties of EUBO. It is favorable to analyze the relationship, e.g., the gap between ELBO and EUBO. In addition, it is unclear the proposed EUBO really minimizes the KL stably. The proposed EUBO is a non-decreasing function? If it is difficult to prove this, at least, they should show the optimization path for ELBO and EUBO in experiments. What is the relationship between ELBO and EUBO? The theoretical analysis in the previous works, e.g., [Li et al., 2016], [Dieng et al., 2016], and [Ji et al., 2019] would be helpful for their further analysis. Furthermore, the performance of the proposed method is crucially depends on the hyperparameter \lambda. If the theoretical guarantee for the selection of \lambda, the usefullness of the proposed method would be more strong. (b) For experiments ・Reproducibility There are few information for experimental settings (e.g., no information about the initial learning-rate). Therefore, there are some problems with reproducibility. What is the value of the initial learning-rate? How many times they iteratively conducted the experiments? How many times the models were trained (what is the value of iteration on SGA)? What is the optimization they used? ・optimization path If it is difficult to guarantee the properties for the proposed EUBO, they should at least show the optimization path based on EUBO and ELBO on training datasets and test ones to confirm that the EUBO is optimizable. It makes strong their claims from the experimental results on class probabilities. ・The reason why the adjusted likelihood did not work well and why r-KL works well I wonder why the adjusted likelihood did not work well and why r-KL works well. I can confirm this phenomenon the authors reported thorough the experiment; however, it would be nice the authors explain their opinion or consideration for this point more in detail. (c) trivial points There are some typos, and it affects the readability of this paper. e.g.: in line 185; stochastic gradient descent (SGA) --> stochastic gradient ascent (SGA) Furthermore, it is helpful for readers to understand the proposed method if they write the algorithm procedure (at least, in appendix). It would be nice how to calculate EUBO (e.g., eq.(8)) concretely with some examples.

Correctness: The claims from the empirical results in this paper seem correct. However, for the empirical methodology, it is not enough to support their own claim. The more information is needed to confirm the properties of EUBO.

Clarity: The claims and organization in this paper are well-written. At least, if the authors wrote details of the experimental settings and fixed the writing (fixed typos), this paper's readability is improved, and we probably understand the merits of this paper more deeply.

Relation to Prior Work: It is necessary to characterize the proposed EUBO through the more theoretical analysis according to the several previous contributions. For example, [Li et al., 2016], [Dieng et al., 2016], and [Ji et al., 2019] may be helpful.

Reproducibility: No

Additional Feedback: All of the comments, suggestions, and questions are in the above sections. ========================= After reading the author response ========================= I would thank the author(s) for their feedback. After reading this, my concern (a) has been addressed somewhat from the new empirical results of the optimization path. However, I still concern about the overall presentation and reproducibility of this paper. Therefore, I keep my score. I think it is necessary to improve the paper presentation drastically and add more detailed information for implementation and experiments to confirm the authors’ claim. Through the polish of presentation (if possible, adding more theoretical investigation with experimental results as the authors showed in their feedback), this paper may turn to be a strong one.


Review 4

Summary and Contributions: *** POST AUTHOR FEEDBACK *** The authors address my points. Therefore, I increase my score. -------- In this work, the authors consider a Bayesian unlearning problem. The authors consider a VI framework to perform an approximate Bayesian unlearning. To address the inaccurate approximation issue, the authors propose two tricks to address this issue.

Strengths: Under the conditional independence (see Lin 91), the authors use the VI framework in the Bayesian unlearning setting and propose new tricks to address the issue in this new setting.

Weaknesses: The presentation is not clear and the empirical evidence can be more convincing if additional experiments are included (see the detailed comments) All experiments seems to be unimodal.

Correctness: The claims seem to be correct. However, due to the VI approximation, it is unclear whether the two tricks play an essential role or not. Additional experiments (see the detailed comments) should be included to clarify this issue.

Clarity: The presentation can be improved. For example, Proposition 1 shows that the evidence upper bound (EUBO) should be optimized in Sec 3.3.1 Footnote 2 for Proposition 1 shows that the reverse KL could be an alternative choice. It may be better to mention Footnote 2 in Sec 3.3.2 so that optimizing the reverse KL makes sense. 

Relation to Prior Work: I am not aware of prior works addressed this unlearning problem.

Reproducibility: Yes

Additional Feedback: Detailed comments: (1) All experiments are about classification and most of them are variational Gaussian approximations.  I would like to see some simple regression experiments. The key point of this experiment to show the proposed tricks work since p(y_x|D_r) can be exactly computed. Therefore, q(y_x|D_r) is not required to compute. For example, consider a Bayesian regression problem with Gaussian likelihood p(y_x|\theta), where p(\theta|D) is a full Gaussian, which can be exactly computed. Let's consider a low-rank Gaussian approximation (e.g., diagonal Gaussian or rank-k-plus-diagional Gaussian ) for q(\theta|D). In this case, p(\theta|D_r) and p(y_x|D_r) can be comptued exactly. Note that \hat{q}_u(\theta|D_r;\lambda) and \hat{q}_v(\theta|D_r;\lambda) should take the same form as q(\theta|D), which is a low-rank Gaussian approximation. The performance of the proposed tricks can be measured by computing the difference between p(y_x|D_r) and  \hat{q}_u(y_x |D_r) or \hat{q}_v(y_x |D_r)  mentioned at Line 209. Note that q(y_x|D_r) in Line 210 is not required to compute. (2) Add experiments when the exact posterior has *many modes*.   The key point of this experiment is to address multimodality, which is closely related to Remark 1. Moreover, all q distributions (q(\theta|D) , q(\theta|D_r) ,  \hat{q}_u(\theta|D_r;\lambda) , \hat{q}_v(\theta|D_r;\lambda)) can take a form of Gaussian or mixture of Gaussians. It also addresses the inaccuracy issue mentioned in Remark 1 since we can add more Gaussian components to reduce the inaccuracy. This experiment empirically supports Remark 1. In this experiment, the ground-truth p(\theta|D) is known. In 2-dimensional cases, p(\theta|D_r) can also be accurately computed by numerical integration due to Eq(2). For example, the exact posterior p(\theta|D) is a *known* mixture of Gaussians distribution. The prior can be computed to as p(\theta) = p(\theta|D) P(D) / P(D|\theta), where P(D) does not depend on \theta and can be dropped in the ELBO. By plugging in this new prior p(\theta) into the ELBO (Line 74), q(\theta|D) can be estimated by optimizing the ELBO. Similarly, q(\theta|D_r) can also be estimated as mentioned at Line 212. Likewise,  \hat{q}_u(\theta|D_r;\lambda) (see Eq (8))  and \hat{q}_v(\theta|D_r;\lambda)  (see Eq (9)) can also be computed. The performance metric at Line 209 can be used to measure the performance of the proposed tricks. This paper becomes more clear if these additional experiments (1) & (2) are included since the p(y_x|D_r) can be accurately computed in these additional experiments.

[Author Response · NeurIPS 2020]

We appreciate the constructive feedbacks from all reviewers, which will be taken into account when revising our paper.

**Reviewer #1**: Our work focuses on the setting of small erased data $\mathcal{D}_e$ (line 28). Since the effect of erasing $\mathcal{D}_e$ from
small training data $\mathcal{D}$ is more noticeable when evaluating our methods, we use small- to moderate-sized $\mathcal{D}$ in our
experiments. To scale to massive datasets, recall that our methods are parsimonious in requiring only $q(\boldsymbol{\theta}|\mathcal{D})$ and $\mathcal{D}_e$
(line 125), i.e., independent of $|\mathcal{D}|$. For example, we use larger-scale sparse GP models for regression on the complex
*Airline dataset* ($|\mathcal{D}| = 2\text{M}$, $|\mathcal{D}_e| = 100\text{K}$) (Hensman et al., 2013): With $\lambda = 0$, EUBO and reverse KL are capable of
unlearning by, respectively, achieving $\text{KL}[\tilde{q}_u(\boldsymbol{\theta}|\mathcal{D}_r) \parallel q(\boldsymbol{\theta}|\mathcal{D}_r)] = 1697.25$ and $\text{KL}[\tilde{q}_v(\boldsymbol{\theta}|\mathcal{D}_r) \parallel q(\boldsymbol{\theta}|\mathcal{D}_r)] = 455.65$
which are smaller than $\text{KL}[q(\boldsymbol{\theta}|\mathcal{D}) \parallel q(\boldsymbol{\theta}|\mathcal{D}_r)] = 4344.09$.

We do not know any unlearning work for approximate Bayesian models; existing works consider MAP and MLE (e.g.,
ridge linear regression, logistic regression). So, there is no suitable existing work for comparison in our experiments.

The disadvantage of overestimating variance in reverse KL can be understood in our study in App. D (referred to in lines
239-247). In Appendix D, we also highlight and discuss practical implications of the main limitation of our approach
when the erased data is informative and only an approximate posterior belief is available (lines 491-501).

To unlearn MCMC, we can re-weight (like importance sampling) MCMC samples by $1/p(\mathcal{D}_e|\boldsymbol{\theta})$ from Eq. (2). We will
consider Laplace approximation for future work. We hope the above results would improve your opinion of our work.

**Reviewer #2**: As you have noticed, our unlearning performance improves when the approximation of the full-data
posterior belief improves due to the challenging constraint of unknown exact full-data posterior belief (lines 125-127).
The Airline experiment described in first paragraph for Reviewer #1 shows the scalability of our methods to a massive
dataset (hence, more expensive model), which will be included in our revised paper. The limitation of our approach is
the dependence on the approximation quality of the posterior belief, which is discussed in Appendix D.

We are not aware of any unlearning work for approximate Bayesian models (i.e., approximate posteriors instead of
MAP or MLE). Therefore, there is no suitable existing work for empirical comparison.

Line 282 is not validated by a flow-based approach but with a multivariate Gaussian approximation (full covariance
matrix) in Appendix E. We will clarify this and address your other comments (e.g., experimental details) in our revision.

**Reviewer #3**: As you have noticed, both EUBO and ELBO minimize the same KL term $\text{KL}[q(\boldsymbol{\theta}|\mathcal{D}_r) \parallel p(\boldsymbol{\theta}|\mathcal{D}_r)]$,
which guarantees their optimal solutions to be the same. We will show an empirical analysis here using the example
of Bayesian linear regression: Fig. (a) below shows both EUBO and ELBO values when minimizing EUBO, while
Fig. (b) shows their values when maximizing ELBO. We can observe that by minimizing EUBO, we maximize ELBO
stably, and vice versa. However, EUBO and ELBO are bounding different quantities, i.e., $\log p(\mathcal{D}_e|\mathcal{D}_r) \neq \log p(\mathcal{D}_r)$.
Therefore, the gap between them is not meaningful. We will discuss the above empirical analysis in our revised paper.

Both EUBO with adjusted likelihood and reverse KL can perform well as they are designed to resolve the issue in
Remark 1 (lines 172-76, 180-81, 186-89). But, EUBO requires a more careful fine-tuning of $\lambda$ to perform well.

We will include more experimental details (which can be extracted from submitted code) and address your other
comments in the revised paper. We hope that the above clarifications would improve your opinion of our work.

**Reviewer #4**: We perform a simple Bayesian regression $y_x = ax^3 + bx^2 + cx + d + \epsilon$ where $a = 2$, $b = -3$, $c = 1$,
$d = 0$, and $\epsilon \sim \mathcal{N}(0, 0.05^2)$. Fig. (c) shows the data. The low-rank approximation of the posterior beliefs are diagonal
Gaussians. Fig. (d) shows samples of $p(y_x|\mathcal{D}_r)$ (exact). Though reverse KL in Fig. (f) and EUBO in Fig. (g) generate
different distributions from the exact $p(y_x|\mathcal{D}_r)$, they resemble $q(y_x|\mathcal{D}_r)$.

Following your suggestion, let $p(\theta|\mathcal{D}) = 0.5\phi(\theta; 0, 1) + 0.5\phi(\theta; 2, 1)$ be a Gaussian mixture (bi-modal) where
$\phi(\theta; \mu, \sigma^2)$ is a Gaussian p.d.f. To easily compare the distributions, let the likelihood of erased data be $p(\mathcal{D}_e|\theta) =$
$1 + \phi(\theta; 2, 1)/\phi(\theta; 0, 1)$. So, $p(\theta|\mathcal{D}_r) = \phi(\theta; 0, 1)$ is a Gaussian by Eq. (2). Supposing the approximate posterior beliefs
are Gaussians, minimizing the KL to the Gaussian mixture $p(\theta|\mathcal{D})$ (or, equivalently, maximizing ELBO) gives $q(\theta|\mathcal{D}) =$
$\phi(\theta; 1.004, 1.390^2)$. Then, given only $q(\theta|\mathcal{D})$ and $p(\mathcal{D}_e|\theta)$, we can compute $\tilde{q}_u(\theta|\mathcal{D}_r; \lambda = 0) = \phi(\theta; 0.060, 1.000^2)$
(minimizing EUBO) and $\tilde{q}_v(\theta|\mathcal{D}_r; \lambda = 0) = \phi(\theta; 0.0618, 1.0184^2)$ (minimizing reverse KL). Hence, EUBO and
reverse KL perform reasonably well (by being close to $p(\theta|\mathcal{D}_r) = \phi(\theta; 0, 1)$) even when $p(\theta|\mathcal{D})$ is bi-modal. We will
include the above results in our revised paper and hope that they would improve your opinion of our work.

(a) Unlearn    (b) Retrain    (c) Data    (d) $p(y_x|\mathcal{D}_r)$    (e) $q(y_x|\mathcal{D}_r)$    (f) $\tilde{q}_v(y_x|\mathcal{D}_r)$    (g) $\tilde{q}_u(y_x|\mathcal{D}_r)$

[Meta-Review · NeurIPS 2020]

This paper is on a fresh and relevant topic that I believe is of increasing interest to the community. The reviewers are all on the fence, with a variety of comments. I have faith that the authors can address many of their concerns in their update for the camera ready and present a manuscript that the NeurIPS community will enjoy discussing.